# THE EFFECTIVENESS OF CURVATURE-BASED REWIRING AND THE ROLE OF HYPERPARAMETERS IN GNNS REVISITED

**Floriano Tori**
Data Analytics Laboratory
Vrije Universiteit Brussel
Floriano.Tori@vub.be

**Vincent Holst**
Data Analytics Laboratory
Vrije Universiteit Brussel
Vincent.Thorge.Holst
@vub.be

**Vincent Ginis**
Data Analytics Laboratory
Vrije Universiteit Brussel
SEAS, Harvard University
Vincent.Ginis@vub.be

## ABSTRACT

Message passing is the dominant paradigm in Graph Neural Networks (GNNs). The efficiency of it, however, can be limited by the topology of the graph. This happens when information is lost during propagation due to being *oversquashed* when travelling through bottlenecks. To remedy this, recent efforts have focused on graph rewiring techniques, which disconnect the input graph originating from the data and the computational graph, on which message passing is performed. A prominent approach for this is to use discrete graph curvature measures, of which several variants have been proposed, to identify and rewire around bottlenecks, facilitating information propagation. While oversquashing has been demonstrated in synthetic datasets, in this work we reevaluate the performance gains that curvature-based rewiring brings to real-world datasets. We show that in these datasets, edges selected during the rewiring process are not in line with theoretical criteria identifying bottlenecks. This implies they do not necessarily oversquash information during message passing. Subsequently, we demonstrate that SOTA accuracies on these datasets are outliers originating from sweeps of hyperparameters—both the ones for training and dedicated ones related to the rewiring algorithm—instead of consistent performance gains. In conclusion, our analysis nuances the effectiveness of curvature-based rewiring in real-world datasets and brings a new perspective on the methods to evaluate GNN improvements.

## 1 INTRODUCTION

Machine learning on graph data, and more specifically Graph Neural Networks (GNNs), has undergone rapid development over the past few years. Both in terms of architecture variations (Wu et al., 2019; Kipf & Welling, 2016; Veličković et al., 2018; Hamilton et al., 2017) as theoretical understanding (Xu et al., 2018; Bronstein et al., 2021; Bodnar et al., 2021a;b). Due to their large flexibility GNNs have been applied in a variety of domains, from physical sciences to knowledge graphs or social sciences (Zhou et al., 2020; Wu et al., 2021). The basis of the *message passing* paradigm of GNNs (Gilmer et al., 2017) rests on the idea of information diffusion where messages, namely the node's feature vector, are propagated along the edges of the graph to their neighbours.

This approach to GNNs has been shown to be very successful, as it combines topological information (the graph) and specific node information (feature vectors) for predictions. However, this paradigm can also suffer from drawbacks. The receptive field of a node, i.e., the number of nodes it can receive information from, depends on the number of layers of the GNN. A low number of layers can cause under-reaching (Sun et al., 2022) where required information does not reach the nodes. On the other hand, if a node's receptive field is too large, this can lead to *oversmoothing* (Li et al., 2018; Oono & Suzuki, 2019), where node representations are homogenised due to subsequent message passing.

Recently, a lot of attention has been drawn to the problem of *oversquashing* (Alon & Yahav, 2021) where structural properties of the graph called bottlenecks cause a loss of information as the messages passing through get too compressed. Research efforts have therefore focused on understanding this

phenomenon (Di Giovanni et al., 2023; Black et al., 2023) as well as on ways to alleviate it. The most pragmatic approach consists of rewiring, i.e., a targeted addition or removal of edges, to reduce the bottlenecks in the graph.

In this work, we investigate the role of graph rewiring to improve performance on node classification tasks. Discrete notions of curvatures on graphs can be used to detect the location of bottlenecks, allowing for the development of algorithmic rewiring methods such as Stochastic Discrete Ricci Flow (Topping et al., 2021). While this point of view was presented first in (Topping et al., 2021), other definitions of curvatures such as Jost and Liu Curvature (Giraldo et al., 2023; Jost & Liu, 2014) or Augmented Forman Curvature (Fesser & Weber, 2023) have been proposed as alternative measures (Bober et al., 2023). Despite that fact that work on synthetic datasets (Di Giovanni et al., 2023; Black et al., 2023) does indicate the occurrence of oversquashing, we here nuance these results on benchmark datasets and reevaluate the performance gains of curvature-based rewiring.

**Contribution**   The goal of our paper is to analyse the effectiveness of curvature-based rewiring to improve performances in non-synthetic graph datasets. This is in line with Tortorella & Micheli (2023) which evaluates rewiring performances on training-free GNNs and shows that rewiring rarely confers a practical benefit for message-passing in those cases. Our work shows that this is the case in general GNNs and offers an explanation: the theoretical motivation for rewiring is not satisfied in most cases when applied to real-world datasets. Additionally, we explain the occurrence of the SOTA results when rewiring by analysing the role of hyperparameter sweeps. We find that these results can be attributed to hyperparameter tuning outliers, as has also been found in other GNN performance gains (Tönshoff et al., 2024). Our work thus calls into question the effectiveness of rewiring for graph datasets and creates a starting point for further investigations on how to evaluate (GNN) improvements and how to bridge theory and experiment beyond synthetic datasets.

**Outline**   In section 2 we introduce the theoretical relation between curvature and bottlenecks in graphs. Section 3 experimentally verifies this condition in standard datasets, where we find that most rewired edges are not necessarily responsible for oversquashing. With this nuanced perspective on the theory underlying the rewiring algorithm, we evaluate the performance gains possible due to rewiring in section 4. When comparing performances of all curvature definitions with the performance obtained when implementing no rewiring at all, we find no consistent (at the level of distributions) improvements over the different datasets.

## 2   GRAPHS, OVERSQUASHING AND CURVATURE

### 2.1   PRELIMINARIES

**Graphs**   We consider an undirected graph $G = (\mathbb{V}, \mathbb{E})$ identified by a set $\mathbb{V}$ of nodes, which are described by a feature vector $\mathbf{x}_i \in \mathbb{R}^{n_0}$ with $i \in \mathbb{V}$, and a set of edges $\mathbb{E} \subset \mathbb{V} \times \mathbb{V}$. The adjacency matrix, which describes the connection of the graph, is denoted by $\mathbf{A}$.

**Graph Neural Networks**   Given the graph $G$ as described as above, we write $\mathbf{h}_i^{(l)}$ the representation of node $i$ at layer $l$, where $\mathbf{h}_i^0 = \mathbf{x_i}$. Given layer dependent, differentiable functions $\phi^l : \mathbb{R}^{n_l} \times \mathbb{R}^{n_l'} \to \mathbb{R}^{n_{l+1}}$ and $\psi^l : \mathbb{R}^{n_l} \times \mathbb{R}^{n_l} \to \mathbb{R}^{n_l'}$, we write the message passing function as

$$\mathbf{h}_i^{(l+1)} = \phi^l \left( \mathbf{h}_i^{(l)}, \sum_{j=1}^{n} \hat{\mathbf{A}}_{ij} \psi^l \left( \mathbf{h}_i^{(l)}, \mathbf{h}_j^{(l)} \right) \right). \tag{1}$$

Here, $\hat{\mathbf{A}}$ denotes the normalised augmented adjacency matrix, i.e. the adjacency matrix $\mathbf{A}$ is augmented by self-loops $\tilde{\mathbf{A}} = \mathbf{A} + \mathbf{I}$ and then normalised by $\tilde{\mathbf{D}} = \mathbf{D} + \mathbf{I}$, where $\mathbf{D}$ denotes the diagonal degree matrix. More precisely, we have $\hat{\mathbf{A}} = \tilde{\mathbf{D}}^{-\frac{1}{2}} \cdot \tilde{\mathbf{A}} \cdot \tilde{\mathbf{D}}^{-\frac{1}{2}}$.

## 2.2 DISCRETE CURVATURE NOTIONS ON GRAPHS

The main idea behind applying discrete curvature notions to detect local bottlenecks in graphs stems from differential geometry. Here, it is well known that the Ricci curvature describes whether two geodesics which start close to each other, either diverge (negative curvature), stay parallel (zero curvature) or converge (positive converge). Prominent examples are the hyperbolic space (negative curvature), Euclidean space (zero curvature) and the sphere (positive curvature). The graph analogues for these spaces are trees, four-cycles and triangles. Discrete curvature notions in essence capture the occurrence of such structures around a given edge. Intuitively, negatively curved edges exhibit more tree-like structures in their local neighbourhoods and are thus prone to oversquash information.

**Discrete curvature notions for graphs**  Curvature notions on graphs depend on topological aspects of the graph with the needed ingredients being the following. For a simple, undirected graph we consider an edge $i \sim j$. We denote by $d_i$ the degree of node $i$ and by $d_j$ the degree of node $j$. The common neighbours of node $i$ and node $j$ are denoted by $\sharp_\triangle(i, j)$. They correspond to the triangles located at edge $i \sim j$. The neighbours of $i$ (resp. $j$) that form a four-cycle based at $i \sim j$ without diagonals inside are denoted by $\sharp_\square^i$ (resp. $\sharp_\square^j$) and the maximum number of four-cycles without diagonals inside that share a common node is denoted by $\gamma_{max}$. We denote by $x \vee y \doteq \max(x, y)$ (resp. $x \wedge y \doteq \min(x, y)$) the maximum (resp. minimum) of two real numbers. In this paper we analyse three main branches of discrete curvatures, with variations within.

First, we consider *Balanced Forman Curvature (BFc)* (Topping et al., 2021) and variations thereof:

- *Balanced Forman Curvature*: For an edge $i \sim j$ we have $BFc(i, j) = 0$ if $\min(d_i, d_j) = 1$ and otherwise

$$BFc(i, j) = \frac{2}{d_i} + \frac{2}{d_j} - 2 + 2\frac{|\sharp_\triangle(i,j)|}{d_i \vee d_j} + \frac{|\sharp_\triangle(i,j)|}{d_i \wedge d_j} + \frac{\left(|\sharp_\square^i| + |\sharp_\square^j|\right)}{\gamma_{max}(d_i \vee d_j)}. \tag{2}$$

- *Balanced Forman Curvature (without four-cycles)*: Determining the number of four-cycles without diagonals inside is a costly computational effort, especially for dense graphs. We therefore analyse the rewiring performance of *BFc* without these four-cycles to evaluate the need of more intensive computations. For an edge $i \sim j$ we have $BFc_3(i, j) = 0$ if $\min(d_i, d_j) = 1$ and otherwise

$$BFc_3(i, j) = \frac{2}{d_i} + \frac{2}{d_j} - 2 + 2\frac{|\sharp_\triangle(i,j)|}{d_i \vee d_j} + \frac{|\sharp_\triangle(i,j)|}{d_i \wedge d_j}. \tag{3}$$

- *Modified Balanced Forman Curvature*: The original code implementation provided in Topping et al. (2021) contained an error in the counting of four-cycles (See Appendix B for more details). We therefore implement this version as well for comparison. For an edge $i \sim j$ we have $BFc_{mod}(i, j) = 0$ if $\min(d_i, d_j) = 1$ and otherwise

$$BFc_{mod} = \frac{2}{d_i} + \frac{2}{d_j} - 2 + 2\frac{|\sharp_\triangle(i,j)|}{d_i \vee d_j} + \frac{|\sharp_\triangle(i,j)|}{d_i \wedge d_j} + \mathcal{O}\left(|\sharp_\square^i| + |\sharp_\square^j|\right). \tag{4}$$

Secondly, we consider *JLc* (Jost and Liu) Curvature(Jost & Liu, 2014). For an edge $i \sim j$ we have, with $s_+ \doteq \max(s, 0)$,

$$JLc(i, j) = -\left(1 - \frac{1}{d_i} - \frac{1}{d_j} - \frac{|\sharp_\triangle(i,j)|}{d_i \wedge d_j}\right)_+ - \left(1 - \frac{1}{d_i} - \frac{1}{d_j} - \frac{|\sharp_\triangle(i,j)|}{d_i \vee d_j}\right)_+ + \frac{|\sharp_\triangle(i,j)|}{d_i \vee d_j}. \tag{5}$$

Finally, we consider the *Augmented Forman Curvature* used for rewiring in (Fesser & Weber, 2023). Originally, the *Forman curvature* was introduced in (Forman, 2003) as a discrete analogue of the Ricci curvature that aims to mimic the Bochner–Weitzenböck decomposition of the Riemannian Laplace operator for quasiconvex cell complexes (compact regular CW complexes which are quasiconvex, i.e. the boundary of two cells can at most consist of one lower-dimensional cell). It was then adapted to graphs (Sreejith et al., 2016) and augmented to also include two dimensional cells such as triangles (Weber et al., 2018; Samal et al., 2018). The *Augmented Forman curvature* comes in two variants.

- A variant where we consider only three-cycle contributions to the curvature. For an edge $i \sim j$ we have

$$\mathcal{AF}_3(i, j) = 4 - d_i - d_j + 3|\sharp_\triangle(i, j)|. \tag{6}$$

- A variant where we also consider the four-cycle contributions to this curvature. It is important to note that, unlike the *Balanced Forman curvature*, the term $\square(i, j)$ in $\mathcal{AF}_4$—as uniquely used in (Fesser & Weber, 2023)—counts all four-cycles located at a given edge $i \sim j$ without obstructions on diagonals inside. For an edge $i \sim j$ we have

$$\mathcal{AF}_4(i, j) = 4 - d_i - d_j + 3|\sharp_\triangle(i, j)| + 2\square(i, j). \tag{7}$$

For all datasets we consider in this paper (see section 3), we computed the curvature distributions of the edges for all different definitions above. These can be found in Appendix A. From these distributions, we note the similar behaviour of *BFc* (and variants thereof) and *JLc*.

**Curvature and oversquashing**  One global measure to identify potential bottlenecks in an undirected graph is given by the Cheeger constant. Small values indicate that the graph can be decomposed into two distinct sets of nodes with only a few edges between those sets. Therefore, the Cheeger constant is only non-zero for connected graphs. Due to the well-known Cheeger inequality, the Cheeger constant can be approximated by the spectral gap, i.e. the first non-zero eigenvalue of the normalised Graph Laplacian. The latter has a lower computational cost. Both notions have been linked to oversquashing. It has been shown (Theorem 6 in (Topping et al., 2021)) that diffusion-based methods (Gasteiger et al., 2019) are limited in their ability to increase the Cheeger constant. In (Di Giovanni et al., 2023), the *symmetric Jacobian obstruction* between two nodes is introduced as a measure of how well information can be exchanged between those nodes. Here, higher values indicate worse information flow and it is shown that the *symmetric Jacobian obstruction* is bounded by the inverse of the Cheeger constant (Corollary 5.6 in (Di Giovanni et al., 2023)).

There have also been attempts to rewire based on a targeted increase of the spectral gap (Karhadkar et al., 2022). However, global measures such as the spectral gap do not convey information about the location of local bottlenecks and thus do not necessarily help to alleviate oversquashing. Here, discrete curvature measures have, allegedly, given us a local and computable metric to estimate this effect based on the graph topology itself. In (Fesser & Weber, 2023), a bound on connections between the neighborhoods of nodes $i$ and $j$ is derived based on $\mathcal{AF}_4 - \mathcal{AF}_4^{\min}$ (Proposition 3.4 in (Fesser & Weber, 2023)). However, this bound is in general not very strict and rather heuristic. In (Nguyen et al., 2023), it is shown edges $i \sim j$ with an *Ollivier (Ricci) Curvature* $\kappa(i, j)$ close to the minimum value of $-2$ cause oversquashing (Propostion 4.4 and Theorem 4.5 in (Nguyen et al., 2023)). From (Topping et al., 2021) we know that $\kappa(i, j) \geq BFc(i, j)$, and through the distribution of $BFc$ in Appendix A we can see that most edges are far away from the lower limit of $-2$ of $BFc$ (and therefore also $\kappa(i, j)$). This will play an important role in the next section.

The fundamental result connecting edges with a very negative *Balanced Forman curvature* to oversquashing is given in Topping et al. (2021), where theorem 4 identifies negatively curved edges as sources of distorted information for a large set of nodes in the local neighbourhood of the given edge.

> **Theorem 4 (Topping et al., 2021)**
>
> Consider a MPNN as in Equation 1. Let $i \sim j$ with $d_i \leq d_j$ and assume that:
>
> 1. $|\nabla\phi_l| \leq \alpha$ and $|\nabla\psi_l| \leq \beta$ with $L \geq 2$ the depth of the MPNN.
> 2. There exists $\delta > 0$ such that $\delta < \frac{1}{\sqrt{(d_i \vee d_j)}}$ ; $\delta < \frac{1}{\gamma_{max}}$ for which $\mathrm{Ric}(i, j) \leq -2 + \delta$
>
> Then there exists $Q_j \subset S_2(i)$ satisfying $|Q_j| > \frac{1}{\delta}$ and for $0 \leq l_0 \leq L - 2$ we have
>
> $$\frac{1}{|Q_j|} \sum_{k \in Q_j} \left| \frac{\partial h_k^{(l_0+2)}}{\partial h_i^{l_0}} \right| < (\alpha\beta)^2 \delta^{\frac{1}{4}}. \tag{8}$$

Theorem 4 above gives a reason to rewire the graph around negatively curved edges (adding triangles or four-cycles which give a positive contribution) as it increases the $\delta$ and therefore softens the bound in equation 8. In (Topping et al., 2021), the set $Q_j$ is explicitly constructed as the neighbours of $j$ that correspond to tree-like structure around the edge $i \sim j$ and the Balanced Forman curvature gives us control over the information flow to these neighbours from node $i$ along the edge $i \sim j$. The idea of the proof is to bound the left-hand side of equation 8 by second powers of the augmented normalised adjacency matrix $(\hat{\mathbf{A}}_{ik})^2$ and then to control these contributions with knowledge about the three- and four-cycles located at edge $i \sim j$ derived from the $\delta$ bound. However, the question arises if the edges selected to be rewired during pre-processing indeed satisfy the conditions (specifically condition 2) of the theorem.

In the next sections, we will explore graph rewiring based on curvature measures in two ways. First, we look at how well Theorem 4 can be applied to benchmark datasets by seeing if edges selected during rewiring are contributing to the bottleneck. Specifically, we look at whether these edges satisfy condition 2. In the second phase, we take a closer look at the performances that different curvature measures deliver. Instead of reporting one-off accuracies originating from parameter sweeps, we take a look at the distribution of accuracies obtained when sweeping parameters.

## 2.3 Graph curvature and rewiring

**Graph rewiring**  In its core idea, graph rewiring techniques encompass recent efforts to decouple the input (true) graph from the graph used for computational tasks. The original message-passing paradigm (Gilmer et al., 2017) for GNNs used the input graph itself. By decoupling this input graph from the graph on which the computations are performed, which can take many forms, one can find alleviations for the oversquashing and -smoothing problem mentioned in the previous section.

One can first look at ways to modify the message passing algorithm on the original graph, for example by sampling only certain nodes to improve scalability (Hamilton et al., 2017), by considering multiple hops during message propagation (Abu-El-Haija et al., 2019; 2020; Zhu et al., 2020) or by changing the definition of a neighbour. This can be done either using shortest path distances (Abboud et al., 2022) or defining importance-based neighbourhoods using random walks (Ying et al., 2018). A more drastic approach consists of changing the graph itself used for propagation, for example by pre-computing unsupervised node embeddings and using neighbourhoods defined by geometric relationships in the resulting latent space for the propagation (Pei et al., 2019), by exchanging the adjacency matrix with a sparsified version of the generalised graph diffusion matrix $\mathbf{S}$, by defining a new weighted and directed graph (Gasteiger et al., 2019), or by propagating messages along a personalised PageRank (Gasteiger et al., 2018).

In this paper, we focus on graph-rewiring techniques as follows: Given a graph $G = (\mathbb{V}, \mathbb{E})$, we construct a new graph $G' = (\mathbb{V}, \mathbb{E}')$ for message-passing. The set $E'$ is constructed based on the discrete curvature measures discussed in the previous section. We treat rewiring as a pre-processing task, similar to methods like SDRF (Topping et al., 2021), BORF (Nguyen et al., 2023), FoSR (Karhadkar et al., 2022), LASER (Barbero et al., 2023), G-RLEF or $\nu$DReW (Gutteridge et al., 2023).

## 3 Benchmark datasets have a lack of sufficiently negatively curved edges

In our experiments, we make exclusive use of the Stochastic Discrete Ricci Flow (SDRF) (Topping et al., 2021) algorithm for rewiring, which works in two steps. First, it selects the most negatively curved edge based on the curvature measure. Around this edge, all edges that can lead to three-cycles and four-cycles are considered, and for each candidate edge, the potential improvement of the curvature is computed. The edge to be added is then selected in a stochastic way, regulated by the temperature parameter $\tau$, where the probability is determined by the improvement the edge brings to the curvature of the original edge. Finally, SDRF also allows, at each iteration, the removal of (very) positively curved edges, determined by the threshold curvature value $C^+$. Although different algorithms have been proposed, we only work with SDRF due to the simplicity of its approach, allowing us to look at the impact of the added edges more clearly.

Looking at the condition from Theorem 4, we now experimentally check if the edges in standard GNN benchmark datasets satisfy the necessary condition 2, which identifies the edge as an 'oversquashing' edge. The number of edges to be rewired is chosen from the hyperparameters given in (Topping et al., 2021). For each edge selected to be rewired (which is the most negative edge) we compute the upper threshold $\delta_{max}(i,j) = BFc(i,j) + 2$. For each of these edges, we then verify if this $\delta_{max}$ determined by the curvature is compatible with the condition

$$\delta < \frac{1}{\sqrt{(d_i \vee d_j)}} \quad \& \quad \delta < \frac{1}{\gamma_{max}} \quad \text{(Condition 2)} \tag{9}$$

which allows the theorem to identify the edge as a bottleneck in the derivation. From our results in Table 1, we see that these conditions on $\delta$ are seldom satisfied by the graphs in the datasets. However, upon examining the derivation of the theorem, we find that the degree-based condition is too stringent. The inequality $0 < \delta < 1/\sqrt{(d_i \vee d_j)}$ is solely used to guarantee that $\delta \leq 1/\sharp_\triangle$, and it is this condition that is subsequently used in the proof. In the second column of Table 1, we display the number of edges that satisfy the actual modified condition required.

$$\delta \leq \frac{1}{\sharp_\triangle} \quad \& \quad \delta < \frac{1}{\gamma_{max}} \quad \text{(Condition 2b)}. \tag{10}$$

Since this bound is looser, we see that more selected edges satisfy the condition 2b, especially when looking at the citation graphs. However, these numbers imply that a part of the edges selected do not satisfy the conditions for Theorem 4, limiting their interpretation as bottlenecks during message passing. Figure 1 shows in which step of SDRF (in % based on maximum number of iterations) the edges that do not satisfy condition 2b are selected to be rewired. Finally, we also note that edges do not satisfy condition 2b due to the three-cycle condition, as well as due to the $\gamma_{max}$ upper limit (as shown by the edges below the dotted line). The figure shows as well that edges that do sat-

Table 1: When an edge is selected by the SDRF algorithm, we verify if this edge satisfies condition 2 of Theorem 4 (Eq. equation 9). Additionally, we check if the edge satisfies a softer, but sufficient, version of condition 2 (Eq. equation 10)

| Dataset | Edges rewired | Cond. 2 (%) | Cond. 2b (%) |
|---|---|---|---|
| Texas | 89 | 0 (0%) | 6 (6.7%) |
| Cornell | 126 | 0 (0%) | 15 (11.90%) |
| Wisconsin | 136 | 0 (0%) | 11 (8.09%) |
| Chameleon | 2441 | 4 (0.16%) | 141 (5.78%) |
| Actor | 1000 | 11 (1.1%) | 237 (23.70%) |
| Squirrel | 787 | 0 (0%) | 34 (4.32%) |
| Cora | 100 | 0 (0%) | 68 (68.0%) |
| Citeseer | 84 | 0 (0%) | 24 (28.57%) |
| Pubmed | 166 | 25 (15.06%) | 116 (69.88%) |
| MUTAG | 3497 | 0 (0%) | 1228 (35.16%) |
| PROTEINS | 50936 | 0 (0%) | 5944(11.67%) |

isfy the condition are sometimes close to the upper bound of $1/\sharp_\triangle$. This again reduces their interpretation as bottlenecks, as the $\delta$ bound on the Jacobian of the features is therefore looser. This temporal information for both types of edges tells us that this phenomenon occurs both at the beginning and the end, indicating that this is not a saturation-type effect.

## 4 HYPERPARAMETER DEPENDENCY OF REWIRING

Graph rewiring algorithms depend on quite some hyperparameters. The first category consists of training and GNNs hyperparameters: learning rate, hidden depth, hidden dimension, weight decay, and dropout. Additionally, there are also the rewiring hyperparameters, which in the case of the SDRF algorithm are: max iterations (i.e. number of edges around which is rewired), temperature $\tau$, and threshold curvature $C^+$. While hyperparameter tuning is an important aspect of optimising models, we argue here that results in accuracy should not be the main judge for a new technique's performance, but one should consider the overall improvement over a wide array of hyperparameters. In this experiment, we perform a parameter sweep over different benchmark graph datasets to evaluate curvature-based graph rewiring with different curvature notions.

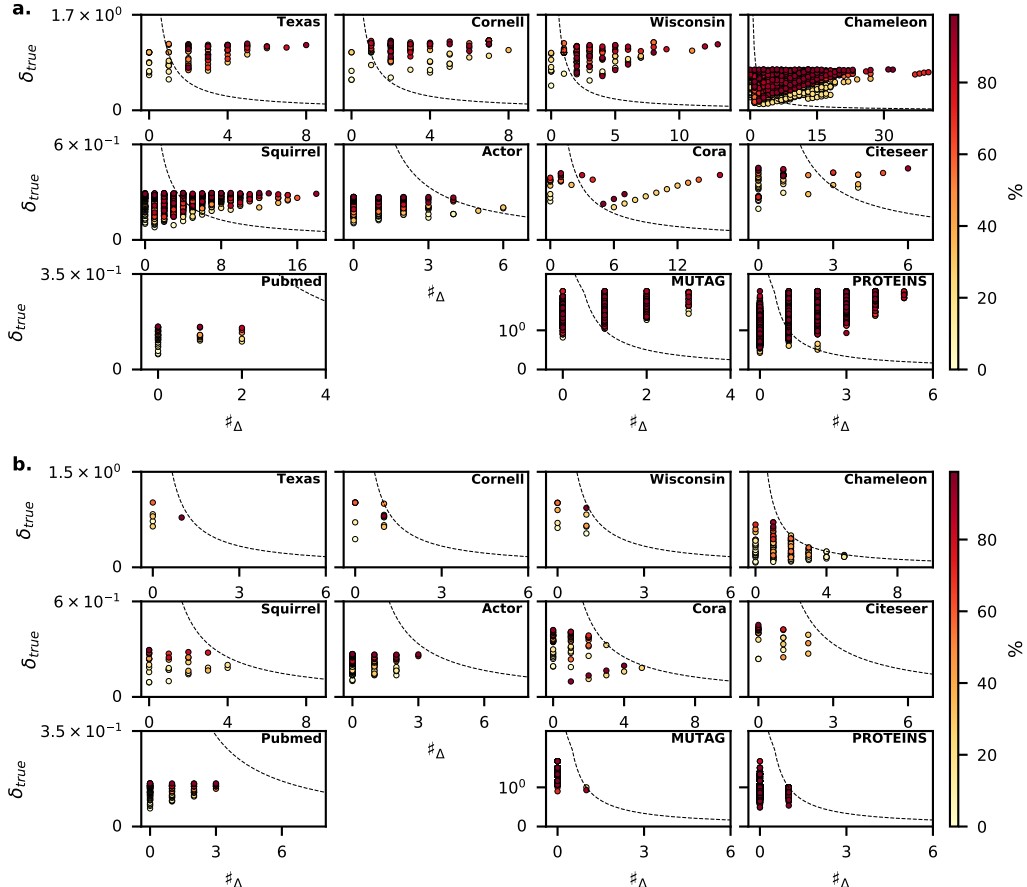

Figure 1: A visualisation of the edges selected during the SDRF rewiring algorithm. **a,** The panels show the edges that do **not** satisfy condition 2b, both due to $\delta > 1/\sharp_\triangle$ (if the edge is situated above the dotted line) or $\delta > 1/\gamma_{max}$ (if the edge is situated below the dotted line). **b,** The panels show the opposite, namely the edges that satisfy condition 2b. This means that the edge is situated below the dotted line and $\delta < 1/\gamma_{max}$. The color code of the edges indicates at which step of the rewiring process (in %) the edge is selected. Dotted line shows $y = 1/\sharp_\triangle$ corresponding to the upper limit in condition 2b.

**Experimental setup** To analyse the impact of different curvatures we ran node-classification tasks. For the node classification tasks we used 9 datasets to evaluate the different curvature notions: Texas, Cornell and Wisconsin from WebKB (University), Chameleon and Squirrel (Rozemberczki et al., 2021), Actor (Tang et al., 2009), Cora (McCallum et al., 2000), Citeseer (Sen et al., 2008) and Pubmed (Namata et al., 2012). For graph classification we used MUTAG and PROTEINS (Morris et al., 2020). For each dataset we only use the largest connected component, extracted with the algorithm provided in `https://github.com/jctops/understanding-oversquashing`. While this implementation does not strictly adhere to the mathematical definition of connected components for directed graphs (it returns neither the strong nor weak component), it was chosen to maintain consistency in the number of nodes and edges, thus facilitating a fair comparison with other studies. Directed graphs were subsequently transformed to undirected.

Our hyperparameter grid is defined as: learning rate: $[0.0001, 0.5555]$, number of layers: $\{1, 2, 3\}$, layer width: $\{16, 32, 64, 128\}$, dropout: $[0.0001, 0.5555]$, weight decay: $[0.0001, 0.9999]$, $C^+$: $[0.2, 21.2]$, $\tau$: $[1, 500]$ and the max number of iterations is dataset dependent, where we take the lower and upper boundary to be $20\%$ above and below the reported best hyperparameters from (Topping et al., 2021). We perform a random-grid search with 800 iterations for all datasets. Our evaluations follows (Gasteiger et al., 2019). Each dataset is split into a development set and a test set with a fixed

seed. The development set is then split 100 times into a validation and training set using the same seeds as in (Topping et al., 2021). Each hyperparameter configuration is then trained and evaluated on each of the 100 training-validation sets. The reported accuracy is then the mean test accuracy over the 100 validation sets. We train the networks with early stopping, where the patience is 100 epochs of no improvement on the validation set. We use a GCN (Kipf & Welling, 2016) for all datasets together with the Adam optimizer (Kingma & Ba, 2014).

**Curvature notions**    The question now arises as to what the role is of different curvature measures when rewiring graphs. For each dataset, we used each discrete curvature notion on graphs discussed in section 2 to rewire the graph in order to assess possible SDRF as extensively as possible.

**Results**    Figure 2 show the smoothed distributions (kernel density estimates), together with box-enplots, of the average accuracy obtained per dataset for each rewiring measure. Additionally, we also look at the top 10% of the reported accuracies given by hyperparameters, which is presented in Table 2. To assess the robustness of the distributions we analysed how well the distribution saturates with an increasing number of added iterations. We look at the evolution of the mean and standard deviation as iterations are added, as well as the Wasserstein distance metric between two distributions which include progressively more iterations. We performed this analysis for all datasets and show it in Appendix E. From these results, we see that our sample of iterations is not unrepresentative of the expected performances and that the obtained distributions saturate, indicating that further runs would not majorly impact the performance distribution.

A first observation is that no curvature measure consistently shifts the distribution of the mean test accuracy away from the None distribution (meaning no rewiring) over all datasets. On some datasets, such as Cornell, Wisconsin or Actor we do notice that $AFc$ based rewiring does provide better performance. However, these curvature definitions do not do this consistently for the other low homophily datasets such as Texas, Squirrel or Chameleon. We also see this behaviour when looking at the average of the top 10% results, where $AFc$ based rewiring does perform better than other variants but with a larger standard deviation. The improvement, taking into account the spread of the distributions is therefore not significant with respect to no rewiring. We can also note the similar performance of less-computational intensive curvature measures ($BFc_3$ and $AFc_3$) in comparison with their more intensive variant ($BFc$ and $AFc_4$).

Table 2: For each dataset we take the top 10% results from the hyperparameter sweeps and compute the average mean test accuracy obtained together with the standard deviation. For some datasets, the top 10% showed almost no variability which resulted in a standard deviation of (almost) 0.

|  | Texas | Cornell | Wisconsin | Chameleon | Cora | Citeseer |
|---|---|---|---|---|---|---|
| None | $59.95 \pm 1.15$ | $53.66 \pm 0.14$ | $54.92 \pm 0.51$ | $40.76 \pm 3.52$ | $58.83 \pm 16.36$ | $58.14 \pm 7.33$ |
| $BFc$ | $59.26 \pm 0.00$ | $53.61 \pm 0.28$ | $54.06 \pm 0.01$ | $34.58 \pm 3.19$ | $28.39 \pm 17.24$ | $35.99 \pm 13.82$ |
| $BFc_3$ | $59.26 \pm 0.00$ | $53.59 \pm 0.12$ | $54.06 \pm 0.01$ | $30.93 \pm 0.10$ | $21.86 \pm 08.75$ | $30.35 \pm 12.03$ |
| $BFc_{mod}$ | $59.26 \pm 0.00$ | $53.68 \pm 0.43$ | $54.91 \pm 1.72$ | $31.60 \pm 2.04$ | $27.73 \pm 13.08$ | $44.16 \pm 12.46$ |
| $JLc$ | $59.26 \pm 0.00$ | $53.57 \pm 0.01$ | $54.06 \pm 0.02$ | $30.91 \pm 0.02$ | $26.83 \pm 13.82$ | $42.48 \pm 7.83$ |
| $AFc_3$ | $59.58 \pm 0.52$ | $54.20 \pm 1.57$ | $56.37 \pm 1.60$ | $36.93 \pm 5.14$ | $59.25 \pm 14.83$ | $60.11 \pm 6.30$ |
| $AFc_4$ | $59.79 \pm 0.54$ | $53.63 \pm 0.10$ | $54.60 \pm 0.80$ | $31.20 \pm 0.62$ | $58.68 \pm 16.10$ | $61.67 \pm 5.43$ |

|  | Pubmed | Actor | Squirrel | MUTAG | PROTEINS |
|---|---|---|---|---|---|
| None | $41.99 \pm 12.58$ | $27.73 \pm 0.02$ | $36.73 \pm 1.96$ | $55.34 \pm 3.67$ | $61.45 \pm 1.49$ |
| $BFc$ | $39.67 \pm 8.30$ | $27.73 \pm 0.01$ | $35.35 \pm 1.00$ | $54.38 \pm 1.71$ | $61.36 \pm 1.23$ |
| $BFc_3$ | $40.97 \pm 12.01$ | $27.73 \pm 0.02$ | $34.66 \pm 0.54$ | $54.45 \pm 2.25$ | $61.16 \pm 0.00$ |
| $BFc_{mod}$ | $41.23 \pm 11.30$ | $27.73 \pm 0.01$ | $34.89 \pm 0.85$ | $54.56 \pm 2.32$ | $61.20 \pm 0.18$ |
| $JLc$ | $39.47 \pm 8.97$ | $27.73 \pm 0.02$ | $34.53 \pm 0.26$ | $54.53 \pm 2.86$ | $61.20 \pm 0.37$ |
| $AFc_3$ | $42.74 \pm 12.91$ | $28.00 \pm 0.93$ | $35.78 \pm 1.83$ | $54.63 \pm 2.91$ | $61.22 \pm 0.45$ |
| $AFc_4$ | $41.40 \pm 12.65$ | $28.14 \pm 1.02$ | $35.64 \pm 1.40$ | $54.54 \pm 2.40$ | $61.27 \pm 1.02$ |

It is also interesting to note that rewiring can also negatively impacted the performance of a dataset, as seen by the larger spread for the citation networks. This could be due to rewiring non-suitable edges which introduces noise in the graph by allowing the communication between nodes that should not. Secondly, when comparing the results from Table 1 with the distribution of $BFc$ in Figure 2, we can see that the datasets with more edges that satisfy condition 2b do not perform better than other (e.g. Pubmed).

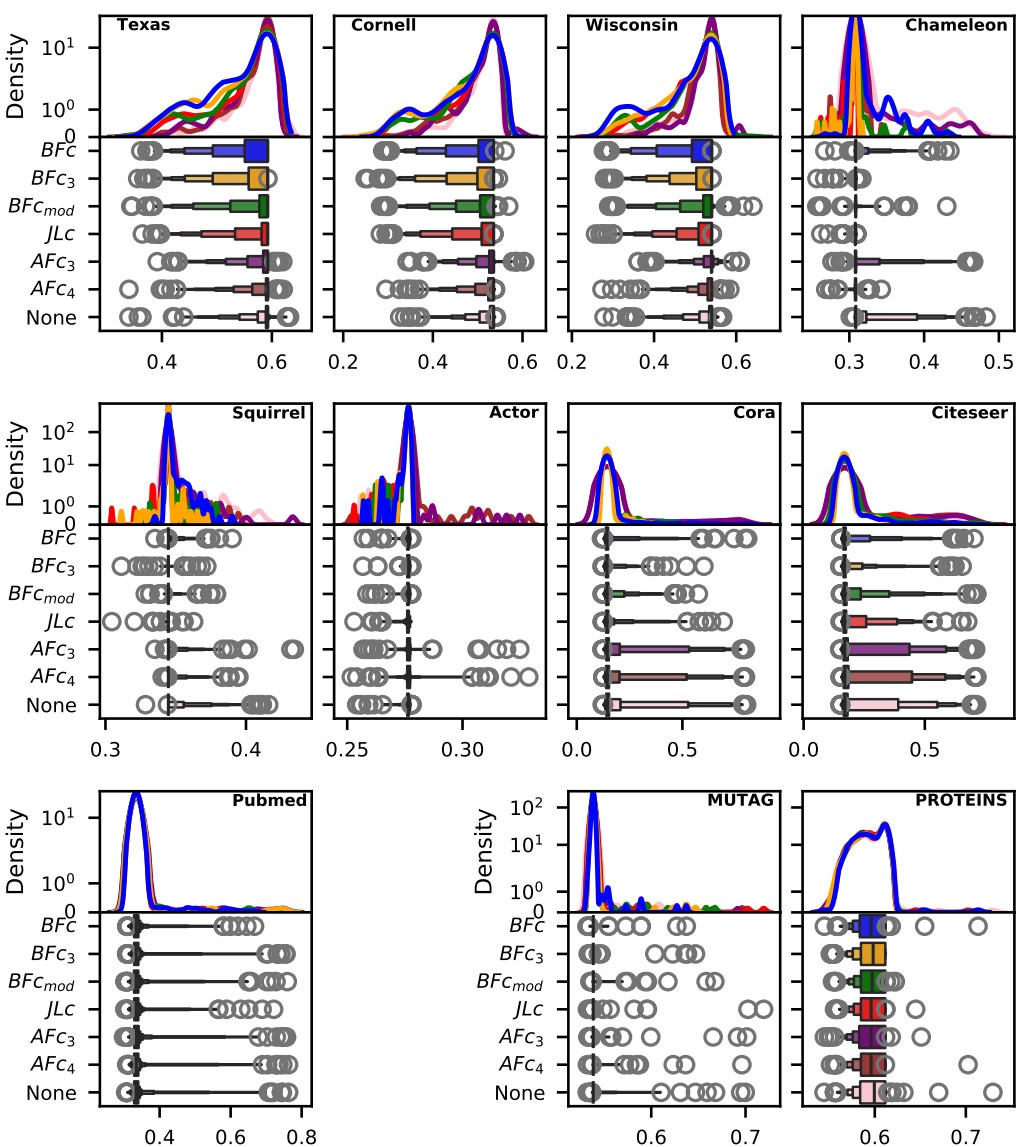

Figure 2: Distribution of mean test accuracy over the sweep of hyperparameters for the different curvature measures and node-classification datasets used. We show boxenplots which first identify the median, then extend boxes outward, each covering half of the remaining data on which outliers (circles) are identifiable. For each dataset we also show the smoothed distribution using kernel density estimates from the seaborn package.

## 5  CONCLUSION

In our work, we have taken a closer look at the effectiveness of graph-rewiring on (large-scale) graph datasets which are commonly used for benchmarking. Our results show that the conditions for over-squashing based on theorems proposed from the literature are not always met when considering these datasets. This implies that the edges selected during rewire do not necessarily cause oversquashing during message-passing and that severe bottlenecks are in fact not present in these datasets. Although one might interpret oversquashing as a continuous phenomenon, these results suggest that it is limited to specific graph topologies.

While graph rewiring can alleviate structural properties of graphs that cause information bottlenecks, thereby mitigating oversquashing, it also interacts intricately with other factors that influence the performance of a GNN. There is an inherent trade-off between reducing oversquashing and enhancing oversmoothing (Nguyen et al., 2023; Fesser & Weber, 2023). Additionally, rewiring can introduce noise, allowing nodes to access information they shouldn't while improving overall information flow.

Moreover, enabling long-range interactions between nodes is only necessary for tasks that depend on those interactions. This explains why citation networks in this paper, despite meeting theoretical bounds better than other datasets (see Table 1), do not show performance improvements with rewiring. This is in line with previous research indicating that these tasks do not rely on long-range interactions (Alon & Yahav, 2021). This dependency is measured by the *homophily* (Pei et al., 2019), and Appendix C shows that citation networks have the highest homophily among the benchmark datasets, further strengthening this observation. For the other datasets, the low homophily indicates the necessity of long-range interactions, aligning with the observation that many edges fail to meet theoretical conditions, as shown in Table 1, and therefore explaining the lack of performance gains for these datasets.

Our analysis is further substantiated by examining the role of hyperparameter sweeping when benchmarking curvature-based graph rewiring methods. We found that most performance gain is due to finding an optimal hyperparameter configuration rather than a structural shift in the performance, as illustrated by the distribution of performance in Figure 2. We argue that future rewiring methods should take into consideration the dependency of their method on hyperparameters, both GNN and rewiring, when evaluating their performance.

**Limitations and future work**  The results presented in this manuscript suggest several directions for future investigation while also facing some limitations. While the distributions shown above might be influenced by the hyperparameter grid used in our random search, focusing on the top 10% of results demonstrates that this influence is likely minimal. Our analysis tested the results for the SDRF algorithm from (Topping et al., 2021), which is the most natural method for curvature-based graph rewiring. It would be interesting to extend this investigation to other rewiring algorithms to validate similar claims. The theoretical analysis of the SDRF bounds indicates that the current bound, $\delta \cdot |\#_\triangle| \leq 1$, could potentially be replaced by a more general bound, $\delta \cdot |\#_\triangle| \leq R$ with $R > 1$, as alluded to in Remark 15 of (Topping et al., 2021). However, this adjustment would further weaken the theoretical bounds established in Theorem 4. The curvature distributions provided in the appendix reveal that the edges are not negatively curved enough to create bottlenecks. Future work could involve developing real-world benchmark datasets that do suffer from bottlenecks to test whether the robust performance gains observed in synthetic data can be replicated. Moreover, exploring the possibility of Theorem-aware rewiring, which involves only rewiring edges that meet theoretical conditions, could be beneficial. This approach should be tested on datasets that genuinely suffer from severe information bottlenecks, as indicated earlier.

ACKNOWLEDGEMENTS

The resources and services used in this work were provided by the VSC (Flemish Supercomputer Center), funded by the Research Foundation - Flanders (FWO) and the Flemish Government. Work at VUB was partially supported by the Research Foundation Flanders under Grants No. G032822N and No. G0K9322N.

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

# A CURVATURE DISTRIBUTIONS

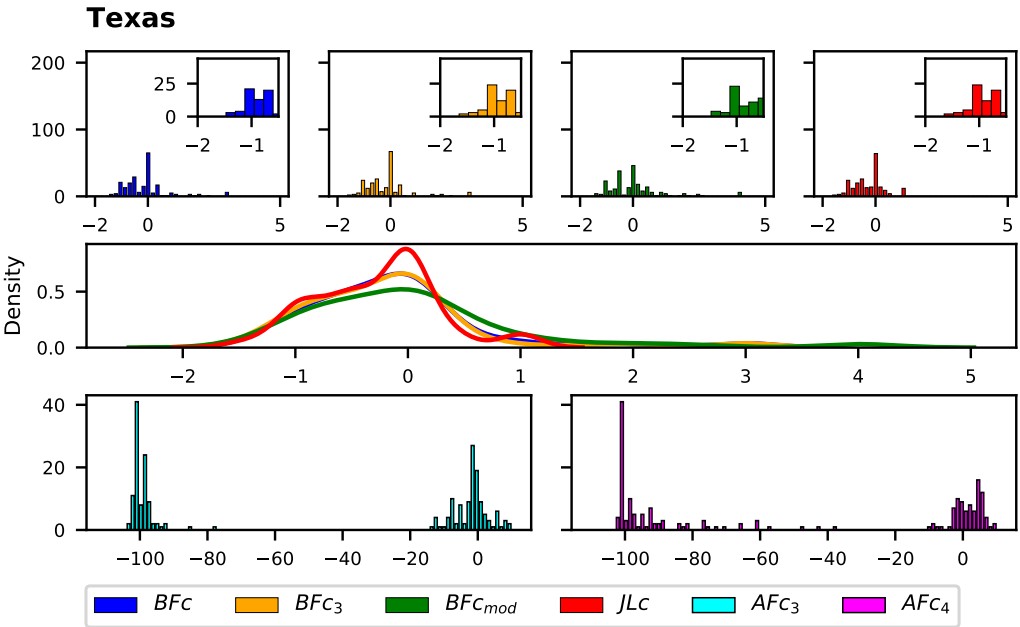

Figure 3: Distribution of curvatures for dataset Texas

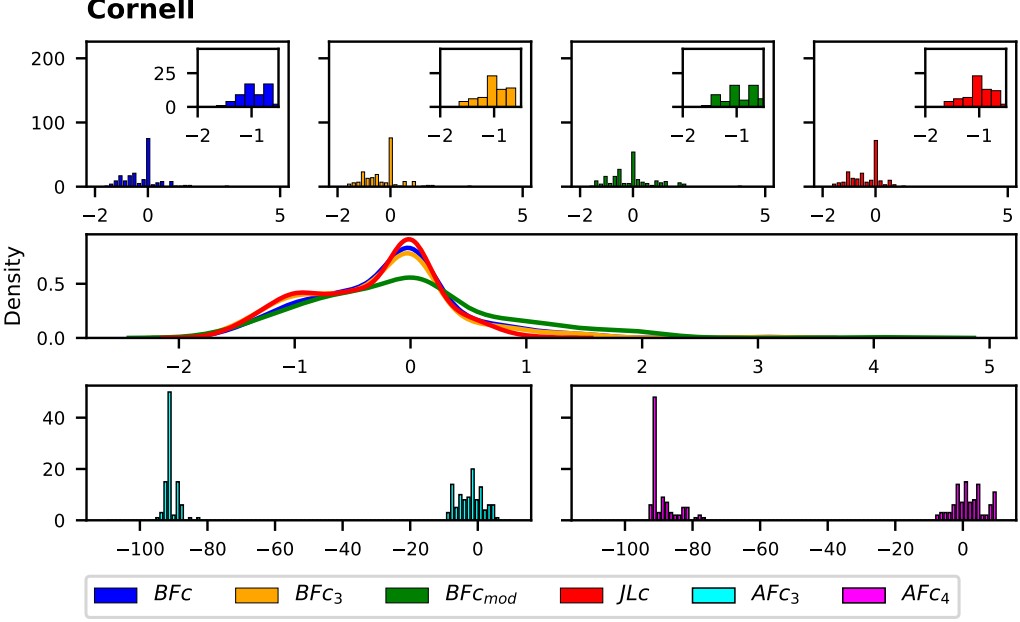

Figure 4: Distribution of curvatures for dataset Cornell

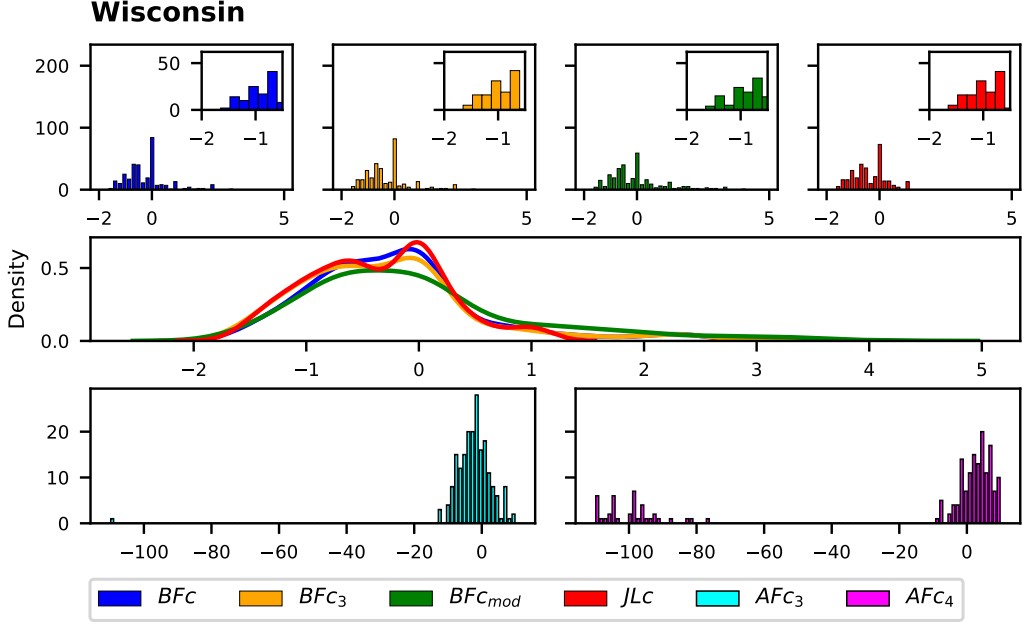

Figure 5: Distribution of curvatures for dataset Wisconsin

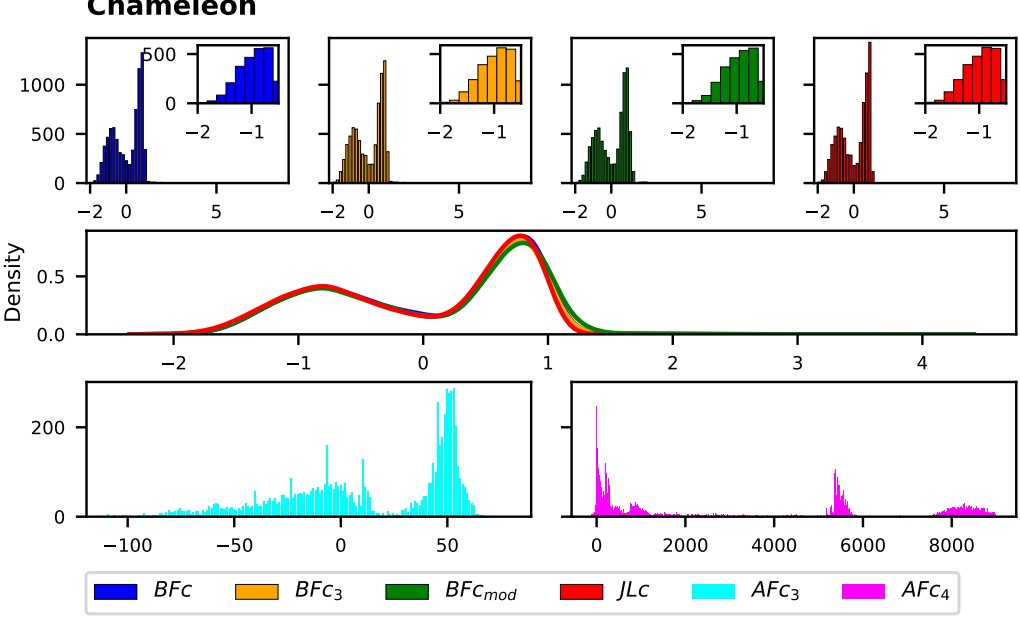

Figure 6: Distribution of curvatures for dataset Chameleon

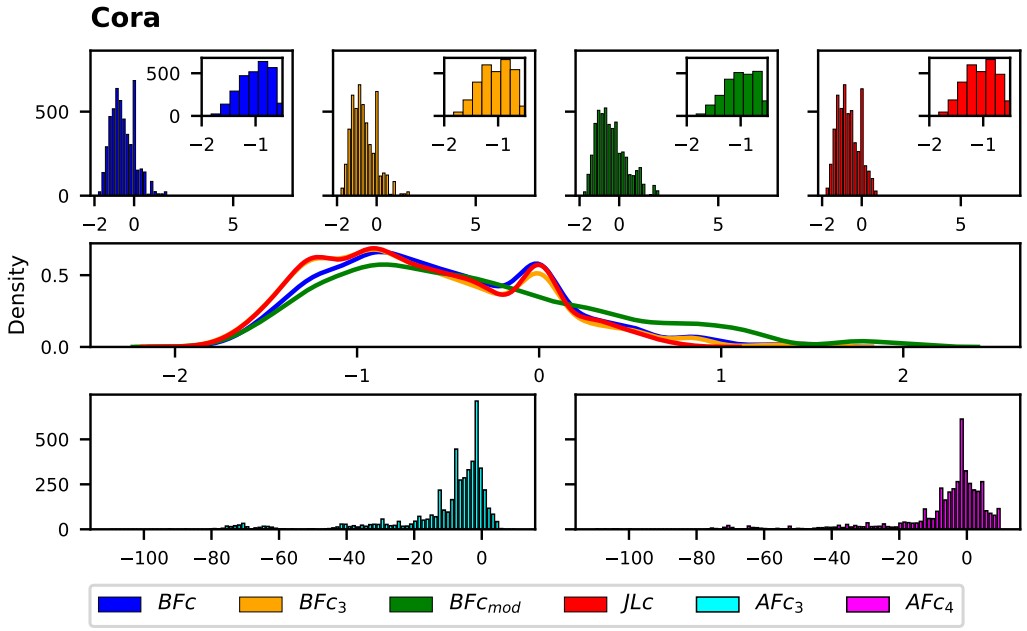

Figure 7: Distribution of curvatures for dataset Cora

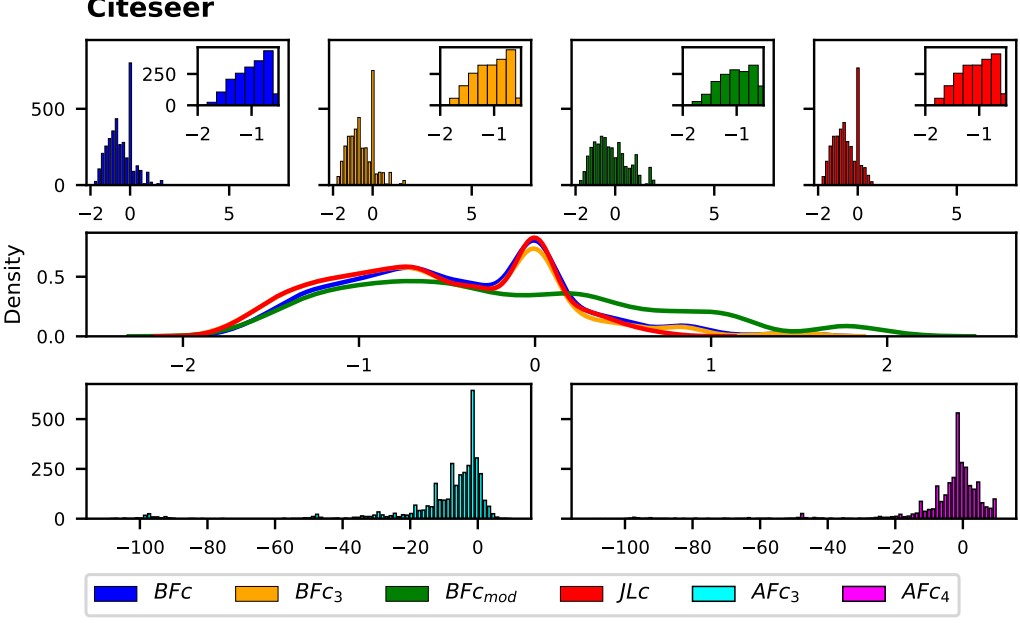

Figure 8: Distribution of curvatures for dataset Citeseer

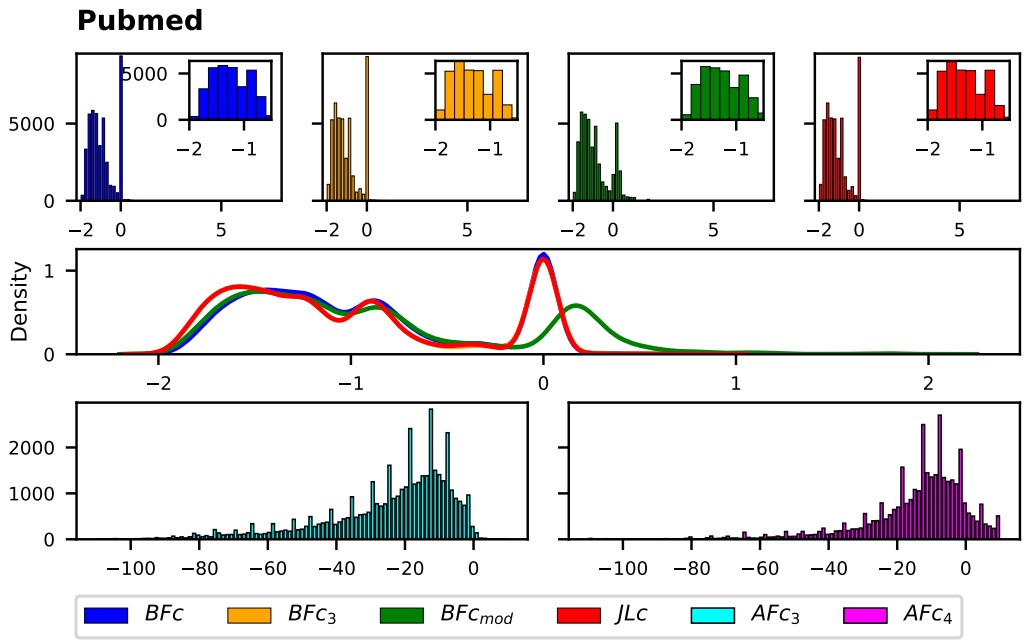

Figure 9: Distribution of curvatures for dataset Pubmed

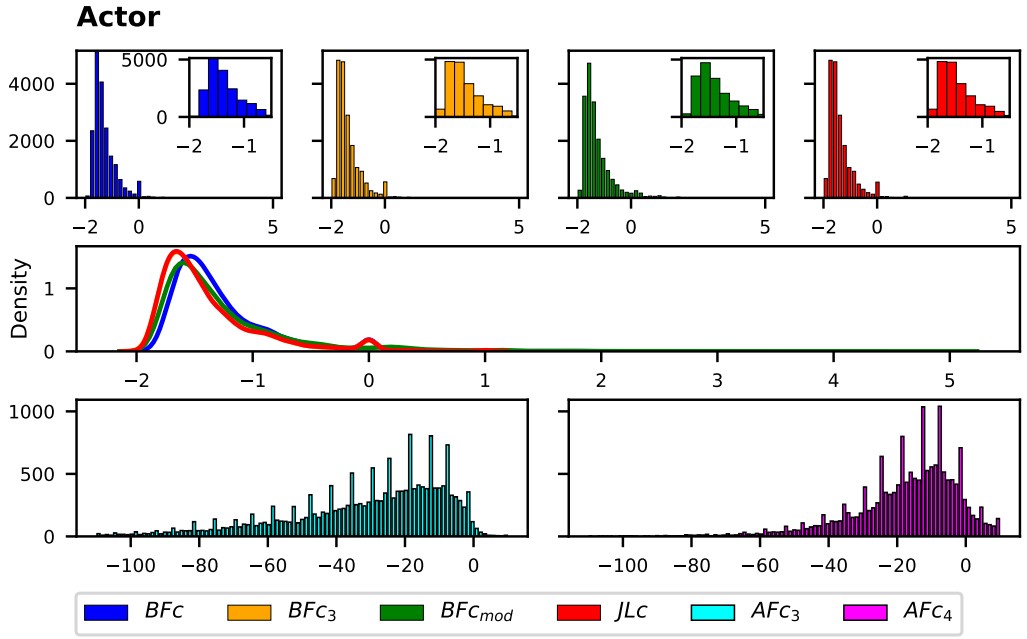

Figure 10: Distribution of curvatures for dataset Actor

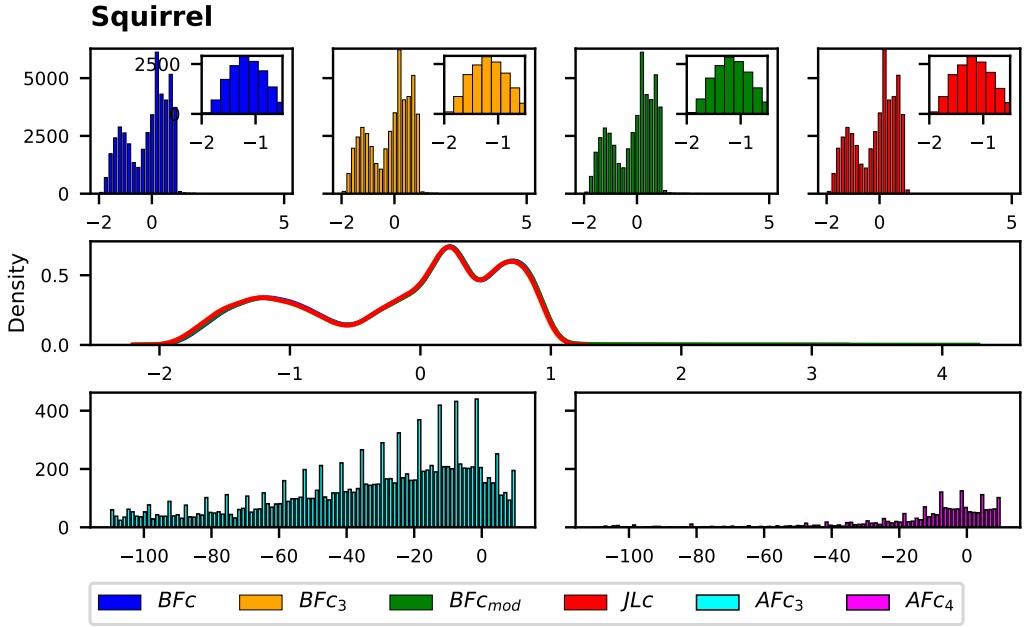

Figure 11: Distribution of curvatures for dataset Squirrel

## B  ON THE IMPLEMENTATION OF THE BALANCED FORMAN CURVATURE

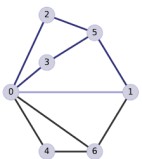

During the setup of our experiment, we noticed that the implementation of the Balanced Forman curvature provided by the authors of Topping et al. (2021) under `https://github.com/jctops/understanding-oversquashing` does not match the theoretical definition presented in their paper (Definition 1 in Topping et al. (2021)). More precisely, the issue for a given edge $i \sim j$ evolves around the terms corresponding to the four-cycle contributions, i.e. $|\sharp_\square^i|$, $|\sharp_\square^j|$ and $\gamma_{max}$. The degree $d_i$ and $d_j$ of the involved nodes and the number of triangles $|\sharp_\triangle(i,j)|$ are calculated correctly. However, even for the sample graph provided in Figure. 3 in Topping et al. (2021) (Figure 12 here) the publicly available implementation produces demonstrably wrong results. The four-cycle contribution is computed as illustrated in the code below with $\mathrm{sharp}_{ij} = |\sharp_\square^i| + |\sharp_\square^j|$, $\mathrm{lambda}_{ij} = \gamma_{\max}$.

Figure 12: Sample graph provided in Figure. 3 in Topping et al. (2021)

If we consider the edge $0 \sim 1$ of the sample graph in Figure 12 and using the definition of the Balanced Forman curvature, Eq. equation 2 we find $BFc(0,1) = 0.10$. In contrast, when using the publicly available code we find $BFc(0,1) = 0.08$.

To control that our computation of $BFc$ was correct we implemented with the *NetworkX* library the set-theoretical definition (evolving around the 1-hop neighbourhoods $S_1(i)$ and $S_1(j)$ of the involved nodes) provided in Definition 1 in Topping et al. (2021). These are

1. $\sharp\Delta(i,j) := S_1(i) \cap S_1(j)$ are the triangles based at $i \sim j$.

2. $\sharp_\square^i(i,j) := \{k \in S_1(i) \backslash S_1(j), k \neq j : \exists w \in (S_1(k) \cap S_1(j)) \backslash S_1(i)\}$ are the neighbours of $i$ forming a 4-cycle based at the edge $i \sim j$ without diagonals inside.

3. $\gamma_{\max}(i,j)$ is the maximal number of four-cycles based at $i \sim j$ traversing a common node

Additionally, we used Remark 10 from Topping et al. (2021) to calculate the four-cycle contribution and in particular $\gamma_{max}$. Our CUDA implementation of *BFc* was then controlled with the *Networkx* implementation. Both our implementations are available in our code.

Listing 1: Code snippet from Topping et al. (2021) to compute the 4-cycle contribution of the *BFc*

```
// 4-cycle contribution
for k in range(N):
        TMP = A[k, j] * (A2[i, k]-A[i, k]) * A[i, j]
        if TMP > 0:
            sharp_ij += 1
            if TMP > lambda_ij:
                lambda_ij = TMP
        TMP = A[i, k] * (A2[k, j]-A[k, j]) * A[i, j]

        if TMP > 0:
            sharp_ij += 1
            if TMP > lambda_ij:
                lambda_ij = TMP

    C[i, j] = ((2 / d_max) + (2 / d_min) - 2 + (2 / d_max + 1 /
        d_min) * A2[i, j] * A[i, j] )
    if lambda_ij > 0:
        C[i, j] += sharp_ij / (d_max * lambda_ij)
```

## C  DATASETS

If the dataset connected disconnected components we report the statistics for the largest connected component selected. The homophily index $\mathcal{H}(G)$ defined by Pei et al. (2019) is defined as

$$\mathcal{H}(G) = \frac{1}{|\mathbb{V}|} \sum_{v \in \mathbb{V}} \frac{\text{Number of } v \text{ 's neighbors who have the same label as } v}{\text{Number of } v \text{ 's neighbors}} . \tag{11}$$

|  | Texas | Cornell | Wisconsin | Chameleon | Squirrel | Actor | Cora | Citeseer | Pubmed |
|---|---|---|---|---|---|---|---|---|---|
| $\mathcal{H}(G)$ | 0.06 | 0.11 | 0.16 | 0.25 | 0.24 | 0.22 | 0.83 | 0.72 | 0.79 |
| Nodes | 135 | 140 | 184 | 832 | 2186 | 4388 | 2485 | 2120 | 19717 |
| Edges | 251 | 219 | 1703 | 12355 | 65224 | 21907 | 5069 | 3679 | 44324 |
| Features | 1703 | 1703 | 1703 | 2323 | 2089 | 931 | 1433 | 3703 | 500 |
| Classes | 5 | 5 | 5 | 5 | 5 | 5 | 7 | 6 | 3 |
| Directed? | Yes | Yes | Yes | Yes | Yes | Yes | No | No | No |

## D  HARDWARE SPECIFICATIONS

Our experiments were performed on a HPC server with nodes containing the following components:

| GPUs per node | GPU memory | processors per node | CPU memory | local disk | network |
|---|---|---|---|---|---|
| 2 x Nvidia A100 | 40 Gb | 2x 16-core AMD EPYC | 256 Gb | 2 TB SSD | EDR-IB |
| 2 x Nvidia Tesla P100 | 16 Gb | 2x 12-core INTEL E5-2650v4 | 256 Gb | 2 TB HDD | 10 Gbps |

# E  SATURATION ANALYSIS OF DISTRIBUTIONS

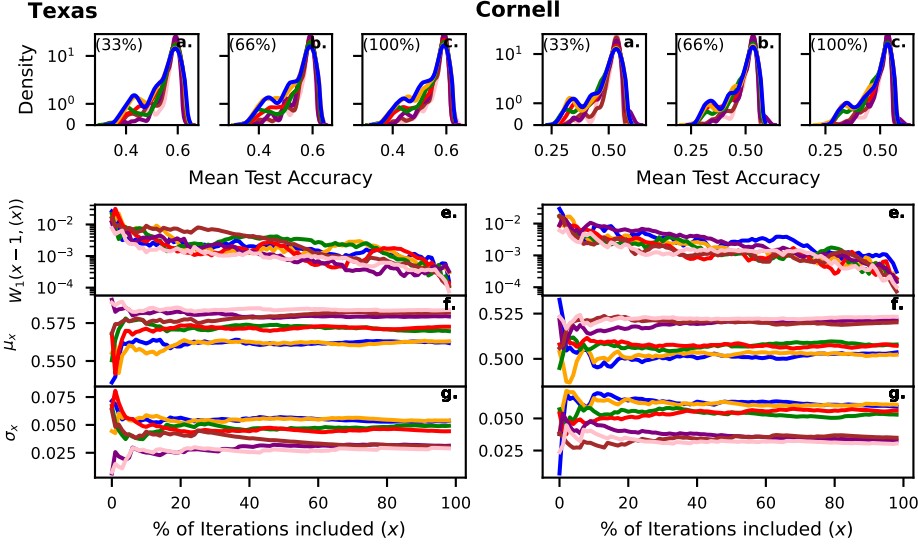

Figure 13: **Saturation analysis for datasets Texas and Cornell (a.-d.)** Distribution of the obtained mean test accuracy when including $33\%$, $66\%$, and $100\%$ of the total hyperparameters runs. **(e.)** Evolution of the Wasserstein distance between two subsequent distributions that include $x\%$ of the total hyperparameter iterations. **(f., g.)** Mean and standard deviation when including $x\%$ of the total iterations.

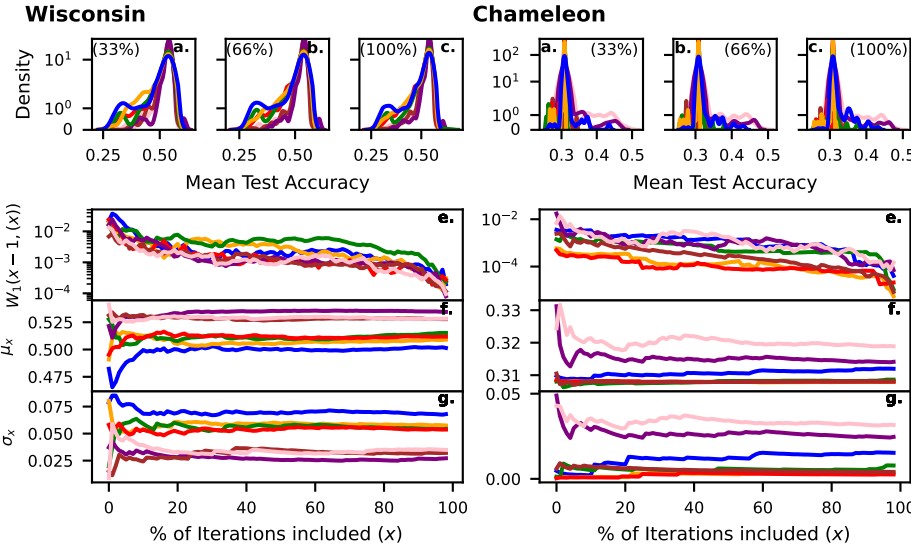

Figure 14: **Saturation analysis for datasets Wisconsin and Chameleon (a.-d.)** Distribution of the obtained mean test accuracy when including $33\%$, $66\%$, and $100\%$ of the total hyperparameters runs. **(e.)** Evolution of the Wasserstein distance between two subsequent distributions that include $x\%$ of the total hyperparameter iterations. **(f., g.)** Mean and standard deviation when including $x\%$ of the total iterations.

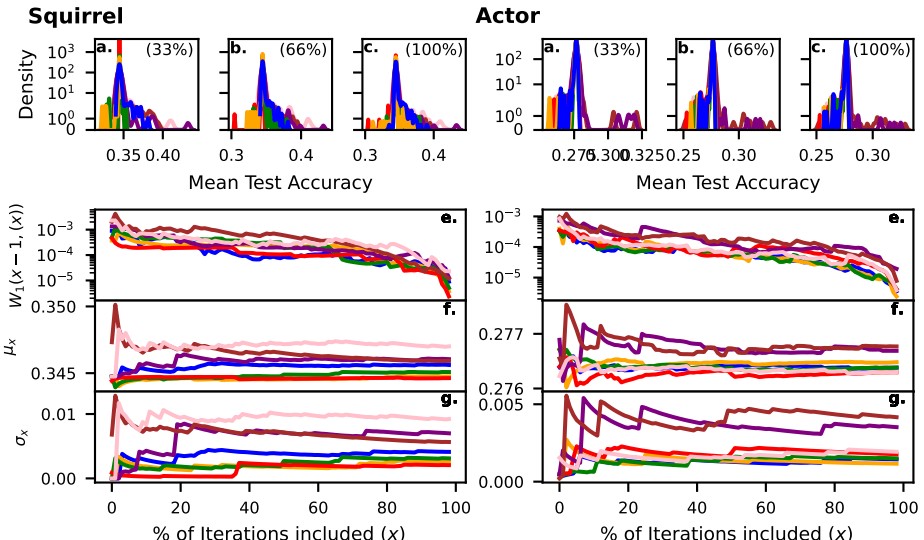

Figure 15: **Saturation analysis for datasets Squirrel and Actor (a.-d.)** Distribution of the obtained mean test accuracy when including 33%, 66%, and 100% of the total hyperparameters runs. **(e.)** Evolution of the Wasserstein distance between two subsequent distributions that include $x\%$ of the total hyperparameter iterations. **(f., g.)** Mean and standard deviation when including $x\%$ of the total iterations.

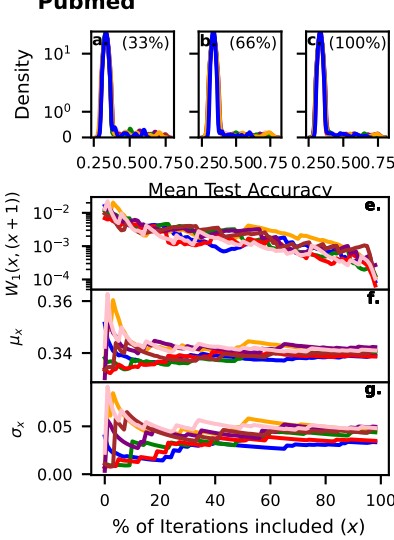

Figure 16: **Saturation analysis for dataset Pubmed (a.-d.)** Distribution of the obtained mean test accuracy when including 33%, 66%, and 100% of the total hyperparameters runs. **(e.)** Evolution of the Wasserstein distance between two subsequent distributions that include $x\%$ of the total hyperparameter iterations. **(f., g.)** Mean and standard deviation when including $x\%$ of the total iterations.

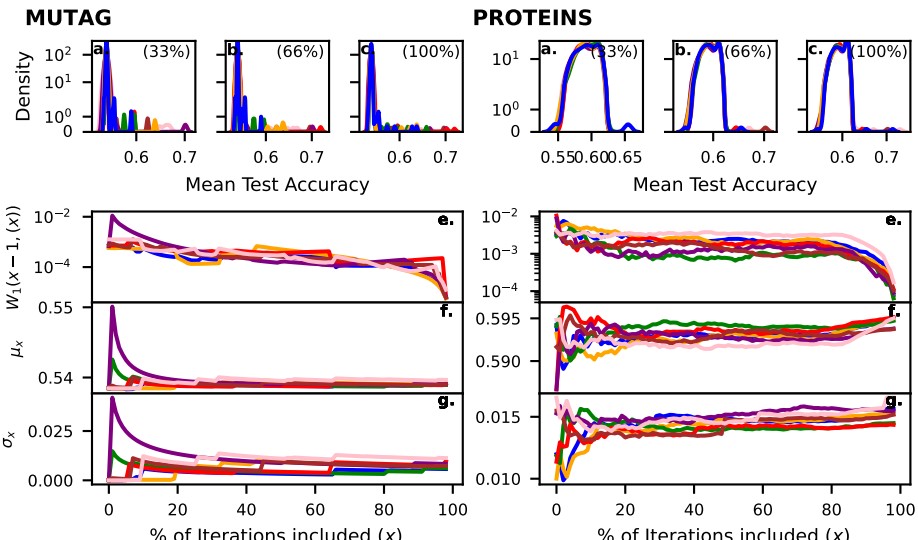

Figure 17: **Saturation analysis for datasets MUTAG and PROTEINS (a.-d.)** Distribution of the obtained mean test accuracy when including $33\%$, $66\%$, and $100\%$ of the total hyperparameters runs. **(e.)** Evolution of the Wasserstein distance between two subsequent distributions that include $x\%$ of the total hyperparameter iterations. **(f., g.)** Mean and standard deviation when including $x\%$ of the total iterations.

