# OpenReview forum: "The Effectiveness of Curvature-Based Rewiring and the Role of Hyperparameters in GNNs Revisited"
_ICLR.cc/2025/Conference — ICLR 2025 Poster_

### Official Review · Reviewer_GwmN · 2024-10-27

**Soundness:** 3
**Presentation:** 2
**Contribution:** 3
**Rating:** 6
**Confidence:** 4

**Summary:**

The paper revisits the effectiveness of curvature-based rewiring in Graph Neural Networks, focusing on its role in alleviating the over-squashing problem. In GNNs, message passing can suffer from information bottlenecks, where messages get compressed, leading to worse downstream performance. Curvature-based rewiring, which involves modifying the graph’s structure to improve information flow, has been proposed as a solution. This paper reevaluates its performance on real-world datasets.

The authors find that in real-world scenarios, the edges selected during the rewiring process do not align with theoretical predictions about bottlenecks, suggesting that the over-squashing issue may not be as prevalent in these datasets. Furthermore, the paper argues that state-of-the-art results from curvature-based rewiring often stem from hyperparameter tuning rather than consistent performance improvements. The study questions the practical benefits of curvature-based rewiring for GNNs and calls for a more nuanced evaluation of GNN improvements.

**Strengths:**

- Clear motivation: The paper has a clear motivation to revisit and critically evaluate the effectiveness of curvature-based rewiring in Graph Neural Networks, and to specifically test whether the theoretical justifications for these methods are genuinely applicable to real-world datasets.
- Extensive experiments: The authors conduct a thorough experimental analysis, testing various curvature measures and examining their effects on node and graph classification tasks. They scrutinize both the theoretical underpinnings and the practical outcomes of rewiring, demonstrating that the edges selected by the rewiring process do not always correspond to bottleneck points. Additionally, they show that state-of-the-art performance is often a result of hyperparameter tuning rather than inherent benefits from rewiring.

**Weaknesses:**

- Presentation: pages 8 and 9 could benefit from some reorganization, for example by better integrating table 2 and figure 2 into the text. Table 2 could also benefit from the best-performing setting being highlighted. I also find Figure 3 hard to read: perhaps it would be better to split the figure in two, i.e. have one figure with the curvature distributions and one with the mean test accuracies.
- More nuanced discussion: while this may be a minor point, I would suggest that the authors include a sentence or two about the role of curvature in graph machine learning more broadly in their discussion section, for example by referring to [1]. While I welcome that the paper pushes back against curvature-based rewiring and consider it good science, this does not mean that curvature is generally not useful for GNNs.
- Long-range datasets: again a minor point, but additional graph-level datasets would further strengthen the paper's message. The authors could, for example, look at Peptides-func and Peptides-struct in the LRGB datasets [2].

[1] Fesser, Lukas, and Melanie Weber. "Effective Structural Encodings via Local Curvature Profiles." The Twelfth International Conference on Learning Representations.

[2] Dwivedi, Vijay Prakash, et al. "Long range graph benchmark." Advances in Neural Information Processing Systems 35 (2022): 22326-22340.

**Questions:**

Could the authors explain why they focus on the theoretical results related to SDRF and not the ones related to BORF, another curvature-based method? Their theoretical results seem less restrictive than what’s presented in the SDRF paper, so I'm wondering what an analogue of Table 1 could look like in this context.

---

> ### Author Response · Authors · 2024-11-21
>
> We would like to thank the reviewer for their feedback and insightful comments. We have responded to them point by point below.
>
> **1. Presentation: pages 8 and 9 could benefit from some reorganization, for example by better integrating table 2 and figure 2 into the text. Table 2 could also benefit from the best-performing setting being highlighted. I also find Figure 3 hard to read: perhaps it would be better to split the figure in two, i.e. have one figure with the curvature distributions and one with the mean test accuracies.
> Concerning Table 2: we will adapt this in the final version of the manuscript together with a reorganisation of the results section of our manuscript.**
>
> Regarding Figure 2: Both the distributions and the boxenplots display the same information in a complementary manner. The boxenplots are helpful in recognizing performance outliers while the distributions provide a more global view of performances over the sweep of hyperparameters. The distributions of the curvature values are included in the Appendix A of our manuscript. They provide a supporting argument for the fact that most edges selected during the rewiring process do not satisfy the conditions from Theorem 4 (as the distributions of curvature values show that not enough edges are negative enough).
>
> **2. More nuanced discussion: while this may be a minor point, I would suggest that the authors include a sentence or two about the role of curvature in graph machine learning more broadly in their discussion section, for example by referring to [1]. While I welcome that the paper pushes back against curvature-based rewiring and consider it good science, this does not mean that curvature is generally not useful for GNNs.**
>
> We agree with the reviewer that curvature in general can be useful for Graph Neural Networks. In the final version, we will revise the discussion sections to include this more nuanced point of view.
>
> **3. Long-range datasets: again a minor point, but additional graph-level datasets would further strengthen the paper's message. The authors could, for example, look at Peptides-func and Peptides-struct in the LRGB datasets [2].**
>
> We agree with the referee that additional robustness checks are valuable to increase the validity of our findings. Computational limitations restrained us from performing the full hyperparameters analysis for the “PascalVOC-SP” dataset from LRGB [2], however, we performed the theorem analysis for this dataset. We find that only 30% of selected edges during rewiring satisfy condition 2b, consistent with our previous findings. This highlights that even in these long-range datasets chosen edges to rewire are not necessarily responsible for oversquashing and (discrete) curvature-based rewiring methods will most likely not yield performance gains on this dataset.
> | Dataset               | Edges Rewired  |   Condition 2  |    Condition 2b      |
> |:---------------------------|:-------------------------:|:---------------------:|:----------------------------:|
> | PascalVOC-SP    |      2 259 482     |      0 (0%)     |   674 596 (29.86 %)
>
> Concerning the datasets used in our work: as the main goal of our work was to re-evaluate the performance of curvature based rewiring we indeed focused on the datasets presented in the original work [1], which were used to assess the validity of the rewiring algorithm. We, therefore, believe that these are then also warranted to re-evaluate its effectiveness and constitute the most transparent approach for this. Our experiments relating to edges satisfying the conditions from Theorem 4 [1] in section 3 then show that these datasets do not possess edges that necessarily oversquash information. Given the impact of the original work [1], we are confident that these results are substantial enough to be presented.

---

> ### Author Response · Authors · 2024-11-21
>
> **4. Could the authors explain why they focus on the theoretical results related to SDRF and not the ones related to BORF, another curvature-based method? Their theoretical results seem less restrictive than what’s presented in the SDRF paper, so I'm wondering what an analogue of Table 1 could look like in this context.**
>
> In the present manuscript, we focus on discrete curvature notions that can be expressed via combinatorial equations and are thus easier to compute than the Ollivier (Ricci) Curvature used for BORF in [3] which involves solving an optimal transport problem. However, thanks to Theorem 2 from [1], we know that the Balanced Forman curvature is a lower bound for the Ollivier (Ricci) Curvature. Therefore, in lines 184-188, our initial submission already included a section in how our results could be applicable to BORF [3]:
>
> ”*In [2], it is shown edges i ∼ j with an Ollivier (Ricci) Curvature κ(i, j) close to the minimum value of −2 cause oversquashing (Propostion 4.4 and Theorem 4.5 in [3]). From [1] we know that κ(i, j) ≥ BF_c(i, j), and through the distribution of BF_c in Appendix A we can see that most edges are far away from the lower limit of −2 of BF_c (and therefore also κ(i, j)).”*
>
> In other words, the BORF method is backed up by theoretical results that identify edges with Ollivier (Ricci) Curvature κ(i, j) close to the minimum value of −2 as bottlenecks and the distributions in Appendix A reveal that this condition is rarely satisfied. Thus, a first proxy for an analogue of Table 1 could be constructed using these distributions.
>
>
> ---
> ---
>
>
> [1] Jake Topping, Francesco Di Giovanni, Benjamin Paul Chamberlain, Xiaowen Dong, and Michael M. Bronstein. Understanding over-squashing and bottlenecks on graphs via curvature. In International Conference on Learning Representations, 2021
>
> [2] Dwivedi, V. P., Rampášek, L., Galkin, M., Parviz, A., Wolf, G., Luu, A. T., & Beaini, D. (2022). Long Range Graph Benchmark. Adv. Neural Inf. Process. Syst. Track Datasets Benchmarks.
>
> [3] Khang Nguyen, Nong Minh Hieu, Vinh Duc Nguyen, Nhat Ho, Stanley Osher, and Tan Minh Nguyen. Revisiting over-smoothing and over-squashing using ollivier-ricci curvature. In International Conference  on Machine Learning, pages 25956–25979. PMLR, 2023

---

> > ### Author Response · Authors · 2024-11-26
> >
> > Dear Reviewer,
> >
> > We hope our response has thoroughly addressed your questions.
> >
> > We would greatly appreciate any further feedback or additional questions you may have.
> >
> > Please let us know.
> >
> > Thank you once again for your thoughtful insights and consideration.
> >
> > Best regards,
> >
> > Authors

---

> > > ### Author Response · Authors · 2024-11-30
> > >
> > > Dear Reviewer,
> > >
> > > We understand that the rebuttal period is a particularly busy time for reviewers, but we would kindly like to ask the reviewer again if our response has thoroughly adressed all their questions. We hope that, if this the case, the reviewer will consider updating their score of our submission.
> > >
> > > Considering the deadlines of the conference we have already modified our submission with the additional experiments (see appendix C), and have adapted the discussion section to better reflect the general use of curvature in Graph Neural Networks as requested by the reviewer.
> > >
> > > Thank you again for your time and questions.
> > >
> > > Best regards,
> > >
> > > The authors

---

> > > > ### Comment · Reviewer_GwmN · 2024-12-02
> > > >
> > > > I thank the authors for their detailed response to my questions and concerns.
> > > >
> > > > While I do support acceptance of the paper, I am hesitant to increase my score to an 8 at this point and prefer to leave it at a 6.

---

> > > > > ### Author Response · Authors · 2024-12-02
> > > > >
> > > > > Dear Reviewer,
> > > > >
> > > > > Thank you for your response. We appreciate the review’s praise and thoughtful suggestions, which we worked diligently to address in full and in detail. Understanding that moving from 6 to 8 is a significant step, we believe the thorough work we’ve done and the absence of remaining criticism warrant reconsideration, and we respectfully ask for it as fellow scientists.
> > > > >
> > > > > Kind regards,
> > > > >
> > > > > The authors

---

### Official Review · Reviewer_UNSN · 2024-10-30

**Soundness:** 2
**Presentation:** 2
**Contribution:** 2
**Rating:** 5
**Confidence:** 4

**Summary:**

This paper reconsiders the effect of rewiring according to curvature on GNN effects. Through a large number of experiments, it points out that the rewiring does not meet the identification criteria of the message-passing bottleneck in the figure, and the effect is not significantly improved under a large number of hyperparameter attempts.

**Strengths:**

1. This article gives a very detailed introduction to curvature rewiring, which is very helpful for readers who are new to the field to understand the work.
2. This paper explains the bottleneck conditions of curvature rewiring from the theoretical point of view and verifies the effect of various methods through experiments, which is very convincing.
3. The paper proves its point through a large number of experiments, which show that the existing methods are ineffective in solving the problem.

**Weaknesses:**

1. The author merely presents a problem, not a solution to it.  The work lacks sufficient integrity.
2. There are some problems with the selection of datasets. For example, MUTAG and PROTEIN, which are themselves molecules and proteins, have biochemical implications. Therefore, performance may not be significantly improved after rewiring. For different areas of graph data, we need to be more profound.
3. The baseline of the experiment needs to be increased. In the task of node classification, the importance of node characteristics is very important. Therefore, the authors need to add an MLP as a baseline. If you can't explain the effectiveness of the edge in this task, then you can't fully explain the problem of rewiring.

Overall, I found this paper a very theoretically solid work. I would like to reconsider my score if my concerns could be addressed.

**Questions:**

1. Can the author provide a comparison of experimental results without using edges? (an MLP with the same number of layers) In general, in a node classification task, node characteristics play a crucial role in the classification effect, often sometimes the role of edges is not obvious.
2. For Other questions please see Weaknesses.

---

> ### Author Response · Authors · 2024-11-21
>
> We would like to thank the reviewer for their time and thoughtful comments. We respond to them point-by-point below
>
> **1. The author merely presents a problem, not a solution to it. The work lacks sufficient integrity.**
>
> Due to the nature of our paper, we agree with the reviewer that our work does not explicitly demonstrate any new methodologies when it comes to rewiring. The focus in our paper is on the performance of curvature-based rewiring on datasets that are used as benchmarks in other rewiring papers instead of synthetic datasets, where bottlenecks can be artificially induced. In doing so we aim to demonstrate that rewiring often fails to meet theoretical conditions, causing the observed improvements to not be theoretically justified. This questions the effectiveness of curvature-based rewiring on improving GNN performances. To explain the results obtained in previous works, we find that the reported improvements may originate from outliers in the hyperparameter sweeps, which can occasionally show high accuracies, instead of consistent performance gains due to rewiring. When comparing the distributions of the results occurring in these sweeps, we show that these outliers occur also when no rewiring has taken place. This observation, together with the previous theoretical analysis, raises questions about the effectiveness of rewiring, aligning with our finding that there is a mismatch between theoretical expectations and experimental results when rewiring edges.
>
> The message of our work is twofold. First, it serves as a re-evaluation of curvature-based rewiring methods, and can importantly influence further development in GNN as we argue that theoretical results and experiments should be more closely checked. Secondly, we advocate for a new and different approach when evaluating methods, where attention is paid to the influence of sweeping hyperparameters.
>
> Additionally, we would greatly appreciate it if the reviewer could kindly elaborate on what they mean by the lack of integrity, so that we can address their concerns more effectively.
>
> **2.There are some problems with the selection of datasets. For example, MUTAG and PROTEIN, which are themselves molecules and proteins, have biochemical implications. Therefore, performance may not be significantly improved after rewiring. For different areas of graph data, we need to be more profound.**
>
> The node-classification datasets and graph-classification tasks (MUTAG and PROTEINS) are all datasets which have been used in prior work [1, 2, 3, 4]. Therefore, we believe that is warranted to re-evaluate the effectiveness of curvature based rewiring methods on these datasets.
>
> However, we agree that certain tasks might not depend on long-range interactions. In lines 495-509 of our initial submission, we elaborate on citation networks (predicting the field of study of a paper might only need the direct neighbors) and also on noise introduced by added edges, allowing distant nodes to communicate even though they should not. In the final manuscript, we would be happy to add a discussion on MUTAG and PROTEINS and their biochemical implications as well.

---

> ### Author Response · Authors · 2024-11-21
>
> **3. The baseline of the experiment needs to be increased. In the task of node classification, the importance of node characteristics is very important. Therefore, the authors need to add an MLP as a baseline. If you can't explain the effectiveness of the edge in this task, then you can't fully explain the problem of rewiring.**
>
> Our study aims to investigate how rewiring impacts the performance of Graph Neural Networks. Since MLPs inherently do not use the graph topology as information for predictions, we do not believe that such a baseline would prove useful in this context. Comparing rewiring within GNNs allows us to “isolate” the effect of rewiring, while comparing against MLPs would fall within a larger evaluation of Graph Neural Networks in general, whose representative power has already been the subject of study [5].
>
> Additionally, we note that standard GNN architectures such as GCNs [6] or GATs [7] are, in their respective works, benchmarked against relevant baselines, providing strong evidence for the use of graph neural networks for these tasks.
>
> ---
> ---
>
>
> [1] Jake Topping, Francesco Di Giovanni, Benjamin Paul Chamberlain, Xiaowen Dong, and Michael M. Bronstein. Understanding over-squashing and bottlenecks on graphs via curvature. In International Conference on Learning Representations, 2021.
>
> [2] Lukas Fesser and Melanie Weber. Mitigating over-smoothing and over-squashing using augmentations of forman-ricci curvature. In The Second Learning on Graphs Conference, 2023
>
> [3] Khang Nguyen, Nong Minh Hieu, Vinh Duc Nguyen, Nhat Ho, Stanley Osher, and Tan Minh Nguyen. Revisiting over-smoothing and over-squashing using ollivier-ricci curvature. In International Conference on Machine Learning, pages 25956–25979. PMLR, 2023.
>
> [4] Federico Barbero, Ameya Velingker, Amin Saberi, Michael M Bronstein, and Francesco Di Giovanni.Locality-aware graph rewiring in gnns. In The Twelfth International Conference on Learning Representations, 2023.
>
> [5] Wang, Xiyuan, and Muhan Zhang. "How powerful are spectral graph neural networks." In International conference on machine learning, pp. 23341-23362. PMLR, 2022.
>
> [6] Thomas N Kipf and Max Welling. Semi-supervised classification with graph convolutional networks. International Conference on Learning Representations (ICLR), 2017.
>
> [7]  Veličković, P., Cucurull, G., Casanova, A., Romero, A., Liò, P., & Bengio, Y. (2018). Graph Attention Networks. Proc. Int. Conf. Learn. Represent

---

> ### Comment · Reviewer_UNSN · 2024-11-24
>
> Thank the authors for their detailed response. While most of my concerns have been answered, I still hold the perspective that this paper presents a problem with new insights, not a new solution, model, or method. As the primary field is selected as "learning on graphs and other geometries & topologies", I expect new learning methodologies rather than re-evaluation (more suitable for the "datasets and benchmarks" primary track). My scores keep the same.

---

> > ### Author Response · Authors · 2024-11-25
> >
> > We would like to thank the reviewer for their continued engagement with our work. We respectfully emphasize that our results represent a significant contribution to the field, as it challenges prevailing assumptions in the field of graph rewiring.
> > Our work addresses fundamental mismatches between theory and practice in curvature-based rewiring, providing insights that prevent reliance on misleading benchmarks and guide future method development. We firmly believe that scientific progress is not solely achieved by proposing new ideas and continually building upon the flawed foundation of previous work and that correcting incorrect results is equally, if not more, critical to ensuring the integrity and advancement of scientific knowledge.
> >
> > We respectfully highlight that our study goes beyond benchmarking by providing important insights into the implementation of graph rewiring techniques and corrects underlying assumptions in the field of graph learning. We hope the reviewer considers this perspective when evaluating the broader impact of our contribution.

---

### Official Review · Reviewer_dc59 · 2024-11-03

**Soundness:** 3
**Presentation:** 2
**Contribution:** 3
**Rating:** 6
**Confidence:** 4

**Summary:**

This paper revisits the effectiveness of curvature-based graph rewiring techniques on real-world datasets. The authors reveal that the identified bottlenecked edges are not in line with theoretical criteria identifying bottlenecks, which thus do not necessarily oversquash the information. Furthermore, the authors demonstrate that the improved accuracies of rewiring techniques on these datasets are outliers originating from sweeps of hyperparameters—both the ones for training and dedicated ones related to the rewiring algorithm—instead of consistent performance gains. They further nuances the effectiveness of curvature-based rewiring in real-world datasets and bring a new perspective on the methods to evaluate GNN improvements

**Strengths:**

1. This paper reveals an important issue in the evaluation of graph rewiring techniques, especially for curvature-based graph rewiring.
2. The theoretical analysis is thorough and solid.

**Weaknesses:**

1. The analysis and empirical study are specific to the curvature-based method (Topping et al., 2021).
2. As the over-squashing issue is highly related to the long-range dependency, the work doesn't include the long-range graph benchmark (Dwivedi et al., 2022), which a bit weakens the study and analysis.
3. The analysis is only on GCN (Kipf & Welling, 2016). It will be more comprehensive to include other widely used MPNNs, e.g., GraphSAGE (Hamilton et al., 2017), GatedGCN (Bresson & Laurent, 2018), GAT (Veličković et al., 2018). This can help better understand the impact of MPNNs on the performance of graph rewiring techniques.









-------
- Dwivedi, V. P., Rampášek, L., Galkin, M., Parviz, A., Wolf, G., Luu, A. T., & Beaini, D. (2022). Long Range Graph Benchmark. _Adv. Neural Inf. Process. Syst. Track Datasets Benchmarks_.
- Bresson, X., & Laurent, T. (2018). Residual Gated Graph ConvNets. In _arXiv:1711.07553_.
- Hamilton, W., Ying, Z., & Leskovec, J. (2017). Inductive Representation Learning on Large Graphs. _Adv. Neural Inf. Process. Syst._, _30_.
- Veličković, P., Cucurull, G., Casanova, A., Romero, A., Liò, P., & Bengio, Y. (2018). Graph Attention Networks. _Proc. Int. Conf. Learn. Represent._

**Questions:**

No further questions beyond weaknesses

---

> ### Author Response · Authors · 2024-11-21
>
> We want to thank the reviewer for their feedback and their questions. We respond point by point here below:
>
> **1. The analysis and empirical study are specific to the curvature-based method (Topping et al., 2021).**
>
> The main focus of our work was to re-analyse the effectiveness of discrete curvature-based rewiring methods. One advantage of those measures, in contrast with other methods such as spectral-based rewiring, is that they can be linked directly to the message passing and local information bottlenecks, as demonstrated in Theorem 4, instead of relying on global measures of the graph that might not reflect the local bottlenecks accurately. Due to Theorem 2 from [1], we know that the Balanced Forman curvature is a lower bound for the Ollivier (Ricci) Curvature. Therefore, in lines 184-188, our initial submission already included a section in how our results could apply to other curvature-base rewiring techniques such as BORF [2]
>
> **2. As the over-squashing issue is highly related to the long-range dependency, the work doesn't include the long-range graph benchmark (Dwivedi et al., 2022), which a bit weakens the study and analysis.**
>
> We agree with the referee that additional robustness checks are valuable to increase the validity of our findings. Computational limitations restrained us from performing the full hyperparameters analysis for the “PascalVOC-SP” dataset from LRGB [3], however, we performed the theorem analysis for this dataset. We find that only 30% of selected edges during rewiring satisfy condition 2b, consistent with our previous findings. This highlights that even in these long-range datasets chosen edges to rewire are not necessarily responsible for oversquashing and (discrete) curvature-based rewiring methods will most likely not yield performance gains on this dataset.
>
> | Dataset               | Edges Rewired  |   Condition 2  |    Condition 2b      |
> |:---------------------------|:-------------------------:|:---------------------:|:----------------------------:|
> | PascalVOC-SP    |      2 259 482     |      0 (0%)     |   674 596 (29.86 %)
>
> Concerning the datasets used in our work: as the main goal of our work was to re-evaluate the performance of curvature based rewiring we indeed focused on the datasets presented in the original work [1], which were used to assess the validity of the rewiring algorithm. We, therefore, believe that these are then also warranted to re-evaluate its effectiveness and constitute the most transparent approach for this. Our experiments relating to edges satisfying the conditions from Theorem 4 [1] in section 3 then show that these datasets do not possess edges that necessarily oversquash information. Given the impact of the original work [1], we are confident that these results are substantial enough to be presented.

---

> ### Author Response · Authors · 2024-11-21
>
> **3. The analysis is only on GCN (Kipf & Welling, 2016). It will be more comprehensive to include other widely used MPNNs, e.g., GraphSAGE (Hamilton et al., 2017), GatedGCN (Bresson & Laurent, 2018), GAT (Veličković et al., 2018). This can help better understand the impact of MPNNs on the performance of graph rewiring techniques.**
>
> We have performed additional analyses to show that our results do not depend on the choice of MPNN architecture (GCN) that was originally used in [1]. To keep the scope of our work aligned with our main message, we have performed, within computational constraints, experiments on three datasets (Texas, Cora and Chameleon) with two additional architectures: GAT [4] and GraphSAGE [5]. The selected datasets are representative for the three categories of node classification datasets used throughout the papers.
> The results are iIn line with what we expected from previous runs. For GraphSAGE, we see that none of the curvature notions consistently improve performances with respect to no rewiring, both on the distribution level as well as on the top 10%. For GAT, we confirm this observation for the Cora and Chameleon datasets. For Texas, we see that AFc-based rewiring does, on occasions, perform well, especially in the top 10% of runs. This is however linked to a large spread in performance as indicated by the standard deviation of the top 10% runs. However, as already mentioned, this performance gain cannot be found for Cora nor Chameleon. We have added these results in Appendix C of our work.
>
>
> ---
> ---
>
>
> [1] Topping, J., Di Giovanni, F., Chamberlain, B. P., Dong, X. & Bronstein, M. M. Understanding over-squashing and bottlenecks on graphs via curvature in International Conference on Learning Representations (2021). 1–3, 7–9
>
> [2] Khang Nguyen, Nong Minh Hieu, Vinh Duc Nguyen, Nhat Ho, Stanley Osher, and Tan Minh Nguyen. Revisiting over-smoothing and over-squashing using ollivier-ricci curvature. In International Conference on Machine Learning, pages 25956–25979. PMLR, 202
>
> [3] Dwivedi, V. P., Rampášek, L., Galkin, M., Parviz, A., Wolf, G., Luu, A. T., & Beaini, D. (2022). Long Range Graph Benchmark. Adv. Neural Inf. Process. Syst. Track Datasets Benchmarks.
>
> [4] Veličković, P., Cucurull, G., Casanova, A., Romero, A., Liò, P., & Bengio, Y. (2018). Graph Attention Networks. Proc. Int. Conf. Learn. Represent
>
> [5] Hamilton, W., Ying, Z., & Leskovec, J. (2017). Inductive Representation Learning on Large Graphs. Adv. Neural Inf. Process. Syst., 30.

---

> > ### Comment · Reviewer_dc59 · 2024-11-25
> >
> > Thank you for the careful rebuttal.
> >
> > I believe the authors have addressed most of my concerns. The challenge to the foundation of curvature-based rewiring techniques is meaningful to the community.
> >
> > I will raise the score to 6 accordingly.

---

> > > ### Author Response · Authors · 2024-11-26
> > >
> > > We appreciate the Reviewer's time and effort in engaging with us and providing valuable, constructive feedback and we are grateful to the Reviewer for improving our scores.
> > >
> > > All the discussed content will be incorporated into the revised manuscript.
> > >
> > > Best regards,
> > >
> > > Authors

---

### Official Review · Reviewer_VNfT · 2024-11-04

**Soundness:** 3
**Presentation:** 3
**Contribution:** 3
**Rating:** 6
**Confidence:** 4

**Summary:**

This paper investigates the effectiveness of curvature-based rewiring in mitigating bottlenecks in graph machine learning tasks. It argues that the theoretical conditions for edges being considered as bottlenecks are not necessarily satisfied for edges being modified in practice. It further argues that the superior performance of some existing methods is likely due to hyperparameter selection rather than systematic improvement.

**Strengths:**

1) The paper takes a careful look at some of the curvature-based methods proposed in the literature and examines whether theoretical conditions match empirical practice. From this perspective, the paper represents a move towards the right direction in evaluation of graph machine learning methods.

2) Detailed description of experimental setup provides helpful guideline for future research in terms of conducting rigorous empirical evaluation. The argument on hyperparameter selection is interesting and points to the importance of a probabilistic view in performance evaluation.

3) The paper is clearly motivated and generally well written. The visualisations are helpful to aid understanding.

**Weaknesses:**

1) As discussed in the paper briefly, I don’t feel the datasets being tested are the most appropriate ones (see Questions below). This makes the findings less surprising and not entirely convincing (although in fairness this is probably a limitation of previous methods as well).

2) Given that this is a paper on empirical validation, experiments should perhaps be done on more than one rewriting method and one single GNN model.

**Questions:**

1) It is unclear whether any dataset in Table 1 would possess bottlenecks that hinder (in particular long-range) interactions that might be necessary for the task (in some sense this is also a limitation of the experiments in Topping et al. 2022), which is one of the main reasons why curvature-based rewiring was proposed in the first place. Therefore the analyses presented in this paper are, albeit interesting and pointing towards the right direction, not entirely surprising. This has been briefly discussed in Section 5, but it might be helpful if the authors can conduct experiments on datasets are may possess long-range interactions, for example the ones described in Dwivedi et al. (https://arxiv.org/abs/2206.08164). Note that the suitability of these datasets are themselves under active debate (see https://arxiv.org/abs/2309.00367), nevertheless they might be more appropriate than the datasets chosen in the paper.

2) The experiments are mostly based on a single rewiring technique and a single GNN model, i.e., the GCN. While this is reasonable starting point, for a more comprehensive evaluation and conclusive evidence, more rewiring methods and GNN models (e.g., GraphSage, ChebNet, or GIN) should be tested. I appreciate the former is recognised as a limitation of the current work, but it makes it less clear how generalisable the findings are.

3) I don’t think homophily is the only factor that determines the long-rangeness of the task. It should depend on the graph topology (e.g., diameter), node features, and the nature of the task as well. Although this is not necessarily the focus of the paper, discussion about this can be made more precise.

---

> ### Author Response · Authors · 2024-11-21
>
> We would like to thank the reviewer for their time and relevant comments. We respond to them point-by-point below
>
> **1. It is unclear whether any dataset in Table 1 would possess bottlenecks that hinder (in particular long-range) interactions that might be necessary for the task (in some sense this is also a limitation of the experiments in Topping et al. 2022), which is one of the main reasons why curvature-based rewiring was proposed in the first place.**
>
> We agree with the referee that additional robustness checks are valuable to increase the validity of our findings. Computational limitations restrained us from performing the full hyperparameters analysis for the “PascalVOC-SP” dataset from LRGB [2], however, we performed the theorem analysis for this dataset. We find that only 30% of selected edges during rewiring satisfy condition 2b, consistent with our previous findings. This highlights that even in these long-range datasets chosen edges to rewire are not necessarily responsible for oversquashing and (discrete) curvature-based rewiring methods will most likely not yield performance gains on this dataset.
>
>
> | Dataset               | Edges Rewired  |   Condition 2  |    Condition 2b      |
> |:---------------------------|:-------------------------:|:---------------------:|:----------------------------:|
> | PascalVOC-SP    |      2 259 482     |      0 (0%)     |   674 596 (29.86 %)
>
> Concerning the datasets used in our work: as the main goal of our work was to re-evaluate the performance of curvature based rewiring we indeed focused on the datasets presented in the original work [1], which were used to assess the validity of the rewiring algorithm. We, therefore, believe that these are then also warranted to re-evaluate its effectiveness and constitute the most transparent approach for this. Our experiments relating to edges satisfying the conditions from Theorem 4 [1] in section 3 then show that these datasets do not possess edges that necessarily oversquash information. Given the impact of the original work [1], we are confident that these results are substantial enough to be presented.
>
>  **2. The experiments are mostly based on a single rewiring technique and a single GNN model, i.e., the GCN. While this is reasonable starting point, for a more comprehensive evaluation and conclusive evidence, more rewiring methods and GNN models (e.g., GraphSage, ChebNet, or GIN) should be tested. I appreciate the former is recognised as a limitation of the current work, but it makes it less clear how generalisable the findings are.**
>
> Concerning the architecture choice: We have performed additional analyses to show that our results do not depend on the choice of GNN architecture (GCN) that was originally used in [1]. To keep the scope of our work aligned with our main message, we have performed, within computational constraints, experiments on three datasets (Texas, Cora and Chameleon) with two additional architectures: GAT [4] and GraphSAGE [5]. The selected datasets are representative for the three categories of node classification datasets used throughout the papers.
>
> The results are in line with what we expected from previous runs. For GraphSAGE we see that none of the curvature notions consistently improve performances with respect to no rewiring, both on the distribution level as well as on the top 10%. For GAT, we confirm this observation for the Cora and Chameleon datasets. For Texas, we see that AFc-based rewiring does, on occasions, perform well, especially in the top 10% of runs. This is however linked to a large spread in performance as indicated by the standard deviation of the top 10% runs. However, as already mentioned, this performance gain cannot be found for Cora nor Chameleon.  We have added these results in Appendix C of our work.
>
> Concerning the rewiring methods: Our goal was to re-analyse the effectiveness of discrete curvature-based rewiring methods. One advantage of those measures, over spectral techniques, is that they can be linked directly to the message passing and local information bottlenecks, as demonstrated in Theorem 4 in ref. [1] instead of relying on global measures of the graph that might not reflect the local bottlenecks accurately. We used SDRF as a rewiring method as it allows us to study the effect of rewiring certain edge, independent of other confounding elements (the identification and rewiring of edges is purely done based on curvature).  According to Theorem 2 from [1], we know that the Balanced Forman curvature is a lower bound for the Ollivier (Ricci) Curvature. Therefore, in lines 184-188, our initial submission already included a section in how our results could apply to other curvature-based rewiring techniques such as BORF [3].

---

> ### Author Response · Authors · 2024-11-21
>
> **3. I don’t think homophily is the only factor that determines the long-rangeness of the task. It should depend on the graph topology (e.g., diameter), node features, and the nature of the task as well. Although this is not necessarily the focus of the paper, discussion about this can be made more precise.**
>
> We agree with the reviewer that the discussion on the long-rangeness of a task should be more elaborate and not only refer to homophily, which is indeed not the only determining factor. In the final version, we will revise the discussion sections to include this more nuanced point of view.
>
>
> ---
> ---
>
>
> [1] Topping, J., Di Giovanni, F., Chamberlain, B. P., Dong, X. & Bronstein, M. M. Understanding over-squashing and bottlenecks on graphs via curvature in International Conference on Learning Representations (2021). 1–3, 7–9
>
> [2] Dwivedi, V. P., Rampášek, L., Galkin, M., Parviz, A., Wolf, G., Luu, A. T., & Beaini, D. (2022). Long Range Graph Benchmark. Adv. Neural Inf. Process. Syst. Track Datasets Benchmarks.
>
> [3] Khang Nguyen, Nong Minh Hieu, Vinh Duc Nguyen, Nhat Ho, Stanley Osher, and Tan Minh Nguyen. Revisiting over-smoothing and over-squashing using ollivier-ricci curvature. In International Conference on Machine Learning, pages 25956–25979. PMLR, 202
>
> [4] Veličković, P., Cucurull, G., Casanova, A., Romero, A., Liò, P., & Bengio, Y. (2018). Graph Attention Networks. Proc. Int. Conf. Learn. Represent
>
> [5] Hamilton, W., Ying, Z., & Leskovec, J. (2017). Inductive Representation Learning on Large Graphs. Adv. Neural Inf. Process. Syst., 30.

---

> > ### Author Response · Authors · 2024-11-26
> >
> > Dear Reviewer,
> >
> > We hope our response has thoroughly addressed your questions.
> >
> > We would greatly appreciate any further feedback or additional questions you may have.
> >
> > Please let us know.
> >
> > Thank you once again for your thoughtful insights and consideration.
> >
> > Best regards,
> >
> > Authors

---

> > > ### Comment · Reviewer_VNfT · 2024-11-27
> > >
> > > I thank the authors for their effort in addressing the points raised in my review. With these further results and discussions I believe the paper is more complete. As I already mentioned in the original review, I believe the attempt made by the authors in this work represents a right direction towards evaluation of graph ML models. I will be raising my rating to 6.

---

### Author Response · Authors · 2024-11-21

We would like to thank the AC for securing reviews with high quality and would like to thank the reviewers for the detailed questions and very thoughtful comments, which helped us highlight better the key contribution of our work.

**Our responses to the Reviewers’ main questions are summarized below.**

* We extended our work to two other graph neural network architectures: GAT [1] and GraphSAGE [2] on three datasets we originally used (Texas, Cora and Chameleon). The datasets are a representative sample of the datasets used throughout our work. These experiments confirmed that performance increases when rewiring are due to hyperparameter tuning instead of rewiring itself.


* We ran our analysis from section 3 as well on the “PascalVOC-SP” dataset from the LRGB [3] datasets which was brought up by different reviewers. We find here as well that only 30% of selected edges during rewiring satisfy condition 2b of Theorem 4 [4], highlighting that even in these long-range datasets edges chosen to rewire around are not responsible for oversquashing.


* We highlighted again how our theoretical results relate to other curvature-based rewiring papers such as BORF [5].

**We welcome any follow-up questions from the reviewers regarding our rebuttal. We hope that, based on our detailed responses, the reviewers will consider increasing their scores if their concerns have been sufficiently addressed.**

---
---

[1] Veličković, P., Cucurull, G., Casanova, A., Romero, A., Liò, P., & Bengio, Y. (2018). Graph Attention Networks. Proc. Int. Conf. Learn. Represent

[2] Hamilton, W., Ying, Z., & Leskovec, J. (2017). Inductive Representation Learning on Large Graphs. Adv. Neural Inf. Process. Syst., 30.

[3] Dwivedi, V. P., Rampášek, L., Galkin, M., Parviz, A., Wolf, G., Luu, A. T., & Beaini, D. (2022). Long Range Graph Benchmark. Adv. Neural Inf. Process. Syst. Track Datasets Benchmarks.

[4] Topping, J., Di Giovanni, F., Chamberlain, B. P., Dong, X. & Bronstein, M. M. Understanding over-squashing and bottlenecks on graphs via curvature in International Conference on Learning Representations (2021). 1–3, 7–9

[5] Khang Nguyen, Nong Minh Hieu, Vinh Duc Nguyen, Nhat Ho, Stanley Osher, and Tan Minh Nguyen. Revisiting over-smoothing and over-squashing using ollivier-ricci curvature. In International Conference on Machine Learning, pages 25956–25979. PMLR, 202

---

### Meta-Review · Area_Chair_RuUT · 2024-12-21

**Metareview:**

Oversquashing has been one of the central challenges in Graph Neural Networks (GNNs). This phenomenon has been demonstrated in synthetic datasets. In the paper, the authors reassess the potential of curvature-based rewiring to enhance the performance of GNNs on real-world datasets. The study investigates the practical application of these techniques, particularly their effectiveness in mitigating the challenges of oversquashing phenomenon in complex and real-world scenarios.

Most of the reviewers agree that the theoretical analyses in the paper are thorough and clear, including a detailed explanation of the bottleneck conditions of curvature rewiring. The experiments are extensive, detailed and convincing, which support the theoretical findings. The presentation of the paper is also clear and easy to follow.

While there was a concern that the paper presents new insights into curvature-based rewiring, rather than proposing a new solution, model, or method, most of the reviewers and I believe that these new insights are useful for the community. As a consequence, I recommend accepting the paper.

The authors are encouraged to incorporate the feedbacks and comments of the reviewers into the revision of their paper.

**Additional Comments On Reviewer Discussion:**

Please refer to the metareview.

---

### Decision · Program_Chairs · 2025-01-22

Accept (Poster)